# Copolymer dielectrics with balanced chain-packing density and surface polarity for high-performance flexible organic electronics

Deyang Ji[1,2], Tao Li [3], Ye Zou[4], Ming Chu[5], Ke Zhou[4], Jinyu Liu[4], Guofeng Tian[6], Zhaoyang Zhang[3], Xu Zhang[3], Liqiang Li [7], Dezhen Wu[6], Huanli Dong[4], Qian Miao[5], Harald Fuchs[1,2] & Wenping Hu[8]

The ever-increasing demand for flexible electronics calls for the development of low-voltage and high-mobility organic thin-film transistors (OTFTs) that can be integrated into emerging display and labeling technologies. Polymer dielectrics with comprehensive and balanced dielectric properties (i.e., a good balance between their insulating characteristics and compatibility with organic semiconductors) are considered particularly important for this end. Here, we introduce a simple but highly efficient strategy to realize this target by using a new type of copolymer as dielectrics. Benefiting from both high chain packing density guaranteeing dielectric properties and surface polarity optimizing molecular packing of organic semiconductors, this rationally designed copolymer dielectric endows flexible OTFTs with high mobility (5.6 cm$^2$ V$^{-1}$ s$^{-1}$), low operating voltage (3 V) and outstanding stability. Further, their applicability in integrated circuits is verified. The excellent device performance shows exciting prospects of this molecular-scale engineered copolymer for the realization of plastic high-performance integrated electronics.

[1] Physikalisches Institut, Westfälische Wilhelms-Universität, Wilhelm-Klemm-Str. 10, 48149 Münster, Germany. [2] Center for Nanotechnology, Heisenbergstr. 11, 48149 Münster, Germany. [3] Shanghai Key Laboratory of Electrical Insulation and Thermal Aging, School of Chemistry and Chemical Engineering, Shanghai Jiao Tong University, 200240 Shanghai, China. [4] Key Laboratory of Organic Solids, Institute of Chemistry, Chinese Academy of Sciences, 100190 Beijing, China. [5] Department of Chemistry, The Chinese University of Hong Kong, New Territories, Shatin, Hong Kong, China. [6] Beijing University of Chemical Technology, 100029 Beijing, China. [7] Advanced Nano-materials Division, Suzhou Institute of Nano-Tech and Nano-Bionics (SINANO), Chinese Academy of Sciences (CAS), 215123 Suzhou, China. [8] Department of Chemistry, Tianjin Key Laboratory of Molecular Optoelectronic Sciences, School of Science, Tianjin University & Collaborative Innovation Center of Chemical Science and Engineering (Tianjin), 300072 Tianjin, China. Correspondence and requests for materials should be addressed to T.L. (email: litao1983@sjtu.edu.cn) or to H.F. (email: fuchsh@uni-muenster.de) or to W.H. (email: huwp@tju.edu.cn)

High-performance flexible organic thin-film transistors (OTFTs) exhibit a great potential in display drivers, smart cards and radio frequency identification tags[1–3]. Wherein polymer gate dielectrics are considered as a key component for the integration of OTFTs into emerging display and labeling technologies due to their intrinsic mechanical flexibility and facile processability for large-area fabrication[4–6]. In addition to the two above-mentioned features, dielectric properties such as good insulating characteristics (i.e., low current leakage and high break-down field) are generally considered one of the pre-requisites for high-performance devices. At the same time, a highly important but often overlooked issue is the compatibility of polymer dielectrics with organic semiconductors (i.e., prefer-able crystalline growth of organic semiconductors on dielectric layer and low interface trap density). However, the key technol-ogy in tuning molecular packing of organic semiconductor on polymeric insulators (e.g., along π-π conjugation direction in conducting channels) is still deficient[7–16]. What is even more concerning is that there is a generally unavoidable compromise between the dielectric properties and molecular packing, which significantly limits the development of high-performance OTFTs.

In the past several decades, a number of polymer dielectrics and their influence on device performances have been investigated[17–23], among which polyimide (PI) is a promising insulator material with comprehensive figures of merit and has shown good feasibility in organic electronics[24–31]. In their pioneer work, Bao and co-workers used screen printing technique to form polyimide layers for the fabrication of OTFTs[26], and it was followed by many reports on a variety of polyimides containing different functional groups for use as dielectrics[27–37]. Despite progress in polyimide synthesis and corresponding device performance, flexible OTFTs and circuits with both high-mobility ($>1 cm^2 V^{-1} s^{-1}$) and low operating voltage (<5 V) remain highly challenging. Recently, we have found that the self-rippled structure of poly (amic acid) (PAA, PI's precursor) strands and the strong polar groups allow more orderly molecular packing on its surface for unprecedented mobility[38]. While com-pared with fully imidized PI, the insulating properties (e.g., current density) of PAA correspondingly decrease about two orders of magnitude due to low chain packing density caused by lack of interaction between the phenyl rings and alicyclic rings[29] and its main body is unstable and easy to degrade after long-time exposure to the air. On the other hand, although high chain packing density could guarantee excellent insulating properties[24], the 2D flat structure of PI strands results in a random orientation of semi-conductor molecules on its surface, and that leads to low mobility[38].

In addition, the high processing temperature (300 °C) of PI is not compatible with low-cost flexible substrates.

Herein, we utilize low-temperature treatment to control the imidization of PAA to prepare a new structure of random copolymer for robust dielectric layers, which not only contains the phenyl rings and alicyclic rings for high chain packing density that improves the insulating property, but also bears the strong polar groups (−COOH/−CONH) favoring molecular packing and charge transport. A good balance of the above two advan-tages achieved by the new structured copolymer dielectric con-tributes to flexible OTFTs with unprecedented performance (mobility up to $5.6 cm^2 V^{-1} s^{-1}$ with operation voltage as low as 3 V for pentacene) and outstanding stability. Furthermore, flex-ible logic circuits such as inverters and oscillators are also suc-cessfully manufactured.

## Results

**The synthesis and characterization of dielectrics.** Figure 1 shows the synthetic route to the copolymer. According to the previously reported procedure[39], the precursor PAA could be easily mass produced (Supplementary Fig. 1) by polymerizing pyromellitic dianhydride (PMDA, $C_{10}H_2O_6$) with 4, 4′-Oxydianiline (ODA, $C_{12}H_{12}N_2O$). Then the PAA was imidized into copolymer films by in-situ annealing treatment at 200 °C (line 2 of Fig. 1). The annealing temperature was chosen considering efficient imidiza-tion (200 °C)[39,40], solvent evaporation (>180 °C, dimethylaceta-mide, DMAc) as well as the compatibility with low-cost flexible substrate (e.g., polyethylene terephthalate (PET), 200 °C is the upper limit of processing temperature). Different from the pre-vious route[24] (line 1 of Fig. 1) to fully (100%) imidized polyimide, our design concept is a combination of the following three points: guaranteeing insulating properties by increasing chain packing density (enhancing the interaction between the phenyl rings and alicyclic rings); retaining a certain number of polar groups (−COOH/−CONH) optimizing molecular packing and charge transport; compatible with solution processing of low-cost flexible devices.

X-ray photoelectron spectroscopy (XPS) was performed to measure the binding energy of C ls, N ls and O ls of PI and the copolymer to analyze chemical composition of their surface. These spectra were deconvolved into typical spectral signatures related to different functional groups[41]. It was found that besides imide groups (O=C–N–C=O, with C 1s, N 1s, and O 1s peaks located at ~288.4, ~400.6, and ~531.7 eV, respectively), there were still carboxylic acid groups (–COOH, with C 1s and O 1s peaks

**Fig. 1** Synthetic route to poly (amic acid) (PAA), polyimide (PI) and copolymer. Preparation of polymer dielectrics

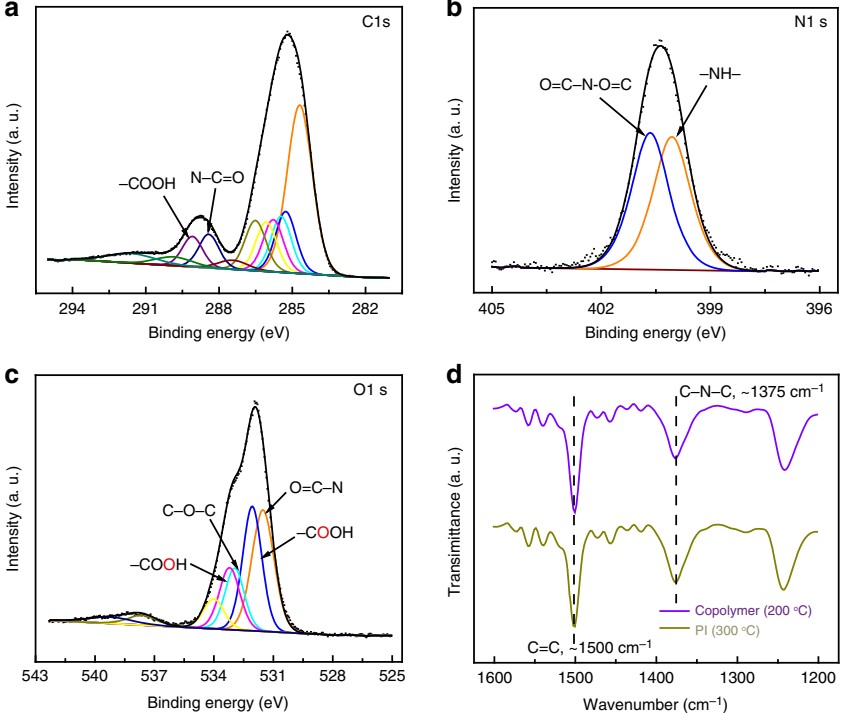

**Fig. 2** XPS and ATR characterizations of copolymer and PI. XPS C ls (**a**), N ls (**b**), and O ls (**c**) spectra of the copolymer and ATR infrared spectroscopy of copolymer and PI (**d**)

located at ~289.1 eV and ~533.3/~532.2 eV, respectively) and nitrogen–hydrogen bonds (–NH–, with N peak located at ~400.0 eV) staying on the surface of the copolymer (Fig. 2a, b, c). Meanwhile only pure imide groups were detected on fully (100%) imidized polyimide surface (Supplementary Fig. 2). The imidization degree was measured by attenuated total reflection infrared spectroscopy (ATR) (Fig. 2d). Theoretically, in the imidization process, stretching vibration of carbon-carbon double bond (C=C, 1500 cm$^{-1}$) of benzene ring keeps unchanged, therefore the ratio of peak intensity between ~1375 (C–N–C) and 1500 cm$^{-1}$ (C=C) is generally used to calculate the degree of the imidization[36-38]. By defining 300 °C annealing resulting in complete imidization (100%), then the imidization degree (200 °C) of copolymer was calculated to be 89%, which further confirmed that polar groups (−COOH/−CONH) existed in the copolymer.

To investigate the dielectric properties of these insulator films, current density and break-down field were tested. A sandwiched device structure of Au/insulator films/indium tin oxide (ITO) was adopted (Fig. 2a inset) and the thickness of each insulator film (around 160 nm) was measured by atomic force microscopy (AFM) (Supplementary Fig. 3). A current density of the copolymer film as low as $10^{-8}$ A cm$^{-2}$ at bias voltage of 5 V was observed (Fig. 3a), which was an order of magnitude lower than that of PAA, indicating that the existence of the phenyl rings and alicyclic rings enhanced the insulating properties. Besides, PAA dielectric layer could only withstand electric field of less than 400 MV m$^{-1}$, while this copolymer film with same thickness exhibited much higher break-down field (about 650 MV m$^{-1}$), which was almost twice that of PAA and very close to fully cross-linked PI (about 660 MV m$^{-1}$) (Fig. 3b). In addition, the copolymer easily formed a compact and uniform film on the flexible substrate as characterized by AFM (Supplementary Fig. 4). Furthermore, it exhibited an excellent transparency of >80% in the visible region (Supplementary Fig. 5).

**The growth mode of pentacene on the dielectrics and characterization of the surface energy**. We used X-ray diffraction (XRD) and 2D grazing-incidence X-ray diffraction (GIXRD) to detect the structural order of the deposited pentacene films. As shown in the XRD patterns, highly ordered pentacene films as formed on the copolymer surface displayed only (00*l*) lattice planes (Fig. 3c). The vertical "Bragg-rod" reflections in the direction of *q*xy (in-plane) combining with the diffraction peak in the direction of *q*z (out-of-plane) of 2D GIXRD revealed an internal edge-on molecular stacking mode of this pentacene film (Fig. 3d). In contrast, the diffraction peak of pentacene thin films on the PI surface was very weak, and the in-plane reflections highly scattered along the Debye rings, indicating that the pentacene molecules displayed edge-on mixing with face-on growth mode on this surface. In addition, the morphology of pentacene films with different thicknesses (4.5, 15 and 50 nm) was further analyzed by AFM. It was clear that pentacene grains exhibited a dendritic structure with a larger grain size of ~1.5 μm (Fig. 3e–g) on the copolymer surface. However, a drastic morphological change of pentacene appeared on the PI surface with small grain size of ~100 nm (Fig. 3h–j). Subsequently, the surface energy of dielectric films and pentacene on the dielectric films was evaluated by measuring the contact angles of water and ethylene glycol[42,43] (Supplementary Fig. 6 and Supplementary Methods). Compared with a large difference in surface energy between PI dielectric (29.4 mJ cm$^{-2}$) and overlying pentacene (50.5 mJ cm$^{-2}$), the copolymer had a surface energy (28.5 mJ cm$^{-2}$) that was similar to pentacene layer (31.4 mJ cm$^{-2}$). The matching of surface energy between the insulator and pentacene is very important, possibly contributing to more efficient transistor channels developed from this interface[14,44]. The details are described in the supporting information and the surface energies of the dielectrics and pentacene films are listed in Supplementary Table 1.

**Flexible organic thin-film transistors**. In order to demonstrate the applicability of copolymer films in flexible devices, pentacene-

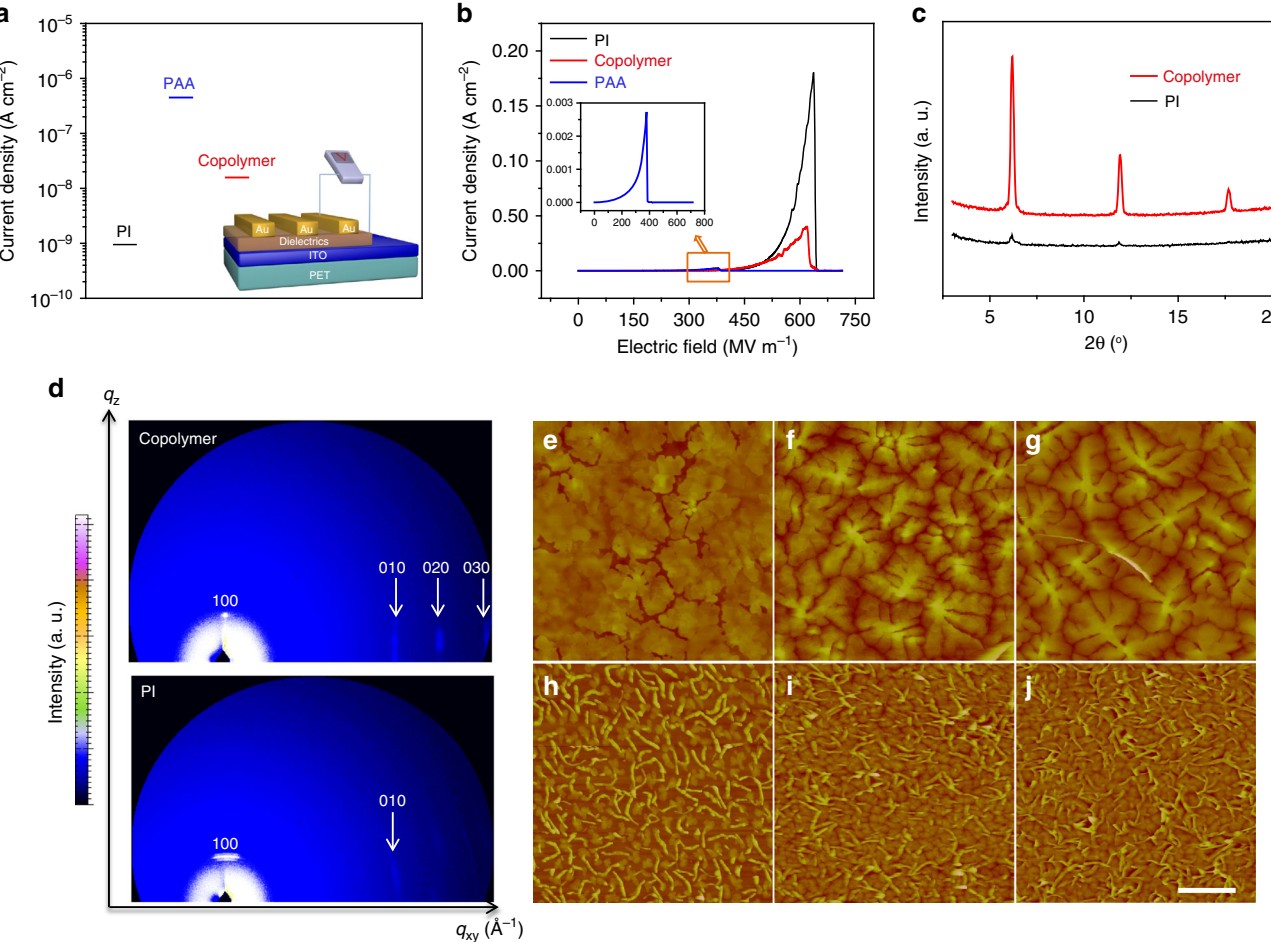

**Fig. 3** The characterizations of dielectric properties and the growth mode of pentacene on the surface of dielectrics. **a** Current density of the dielectric layers at the bias voltage of 5 V. Inset, an Au/dielectrics (160 nm)/ITO sandwiched device structure for tests. **b** Current density as a function of electric field. **c** XRD patterns of pentacene films (50 nm) grown on copolymer and PI surfaces. **d** 2D GIXRD patterns of pentacene films (50 nm) on the surface of copolymer and PI. AFM images of pentacene films grown on dielectric substrates with different thicknesses (4.5, 15, and 50 nm). **e**–**g** on copolymer surface; **h**–**j** on PI surface. Scale bar, 1 μm

based OTFTs with this copolymer as gate dielectrics and vapor-deposited layers of 20 nm Au as source and drain electrodes were fabricated on PET substrates. Supplementary Fig. 7 shows specific capacitance as a function of frequency from 20 Hz to 100 kHz of Au/copolymer(160 nm)/ITO in the air. The capacitance per unit area ($C_i$) of this thin film was measured to be about 20 nF cm$^{-2}$ (at 20 Hz). To more accurately calculate the dielectric constant, capacitance values of the copolymers with various thicknesses were tested (from 20 Hz to 100 kHz) both in the air (under relative humidity of ~80–90%) and in the vacuum (Supplementary Fig. 8). It was clear that there was little difference of capacitance in the air and in the vacuum, even in the low frequency region, suggesting that the humidity had little effect on the copolymer film. As a result, the dielectric constant of this copolymer can be calculated around 4. Figure 4a shows the process to fabricate large-area flexible OTFT arrays, wherein a scheme of the OTFT device structure with channel widths and lengths of 240 and 30 μm is exhibited. All the devices exhibited mobility values above 2 cm$^2$ V$^{-1}$ s$^{-1}$ (see an example in Fig. 4b). The highest mobility calculated from saturation region (Fig. 4c) reached 5.6 cm$^2$ V$^{-1}$ s$^{-1}$ (at 20 Hz), with an on/off current ratio of 1.4 × 10$^6$, a threshold voltage of 0.42 V and a sub-threshold swing of 220 mV dec$^{-1}$. This mobility value was almost ten times higher than the best performance of our previous OTFTs (0.55 cm$^2$ V$^{-1}$ s$^{-1}$) using PI dielectrics[24]. From Fig. 4c, the gate current was smaller than the drain current by more than five orders

of magnitude, which further confirmed the high insulating property of this copolymer film. In addition, the subthreshold swing showed a low interface trap density[45] of 3.7 × 10$^{11}$ cm$^{-2}$ eV$^{-1}$. As compared to the one-order and two-order of magnitude higher trap density in PI (160 nm) and SiO$_2$ (50 and 300 nm), respectively (Supplementary Fig. 9), the low interface trap density of copolymer indicated excellent dielectric-semiconductor interface quality and also demonstrated only a low gate voltage was required to attract holes to fill the charge trap states before accumulation occurring during operation of the OTFT[46]. By using 160 nm-thick copolymer dielectric, the operating voltage was reduced to be as low as 3 V, which was more than an order of magnitude smaller compared with previous reports, representing a big step forward towards practical application of polyimide-based OTFTs. What's more, the devices were stable in the atmosphere environment with little influence by the moisture or other impurities in air, as indicated by their almost the same performance in the air and in the vacuum (Supplementary Fig. 10). The robustness of the devices could be ascribed to the close interaction between the copolymer and pentacene keeping their interface from being affected by ambient conditions[38]. A typical output characteristic of this OTFT is displayed in Supplementary Fig. 11, showing the expected gate modulation of the drain current in both linear and saturation regimes. Forward and backward characteristics of the OTFT were measured (Supplementary Fig. 12a, b), and the hysteresis effect was negligible, which was

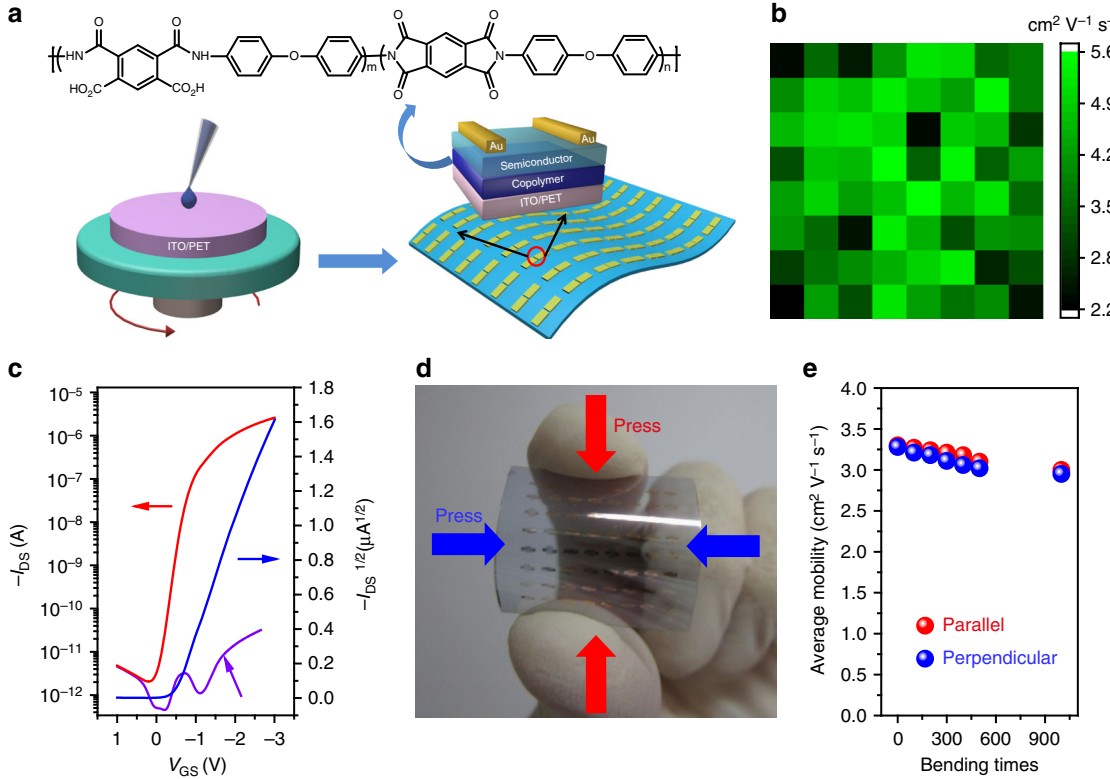

**Fig. 4** Device structure and performance. **a** The preparation process for large-area flexible OTFT arrays. **b** Distribution of device mobility. **c** Typical transfer curve of the OTFT with 50 nm pentacene and a channel dimension of $W = 240\ \mu m$, $L = 30\ \mu m$. The gate current as a function of gate-source voltage is shown in purple. **d** A photograph of flexible devices for the test of bending effect. **e** Plots of mobility versus bending times on PET substrate based on copolymer insulating layers

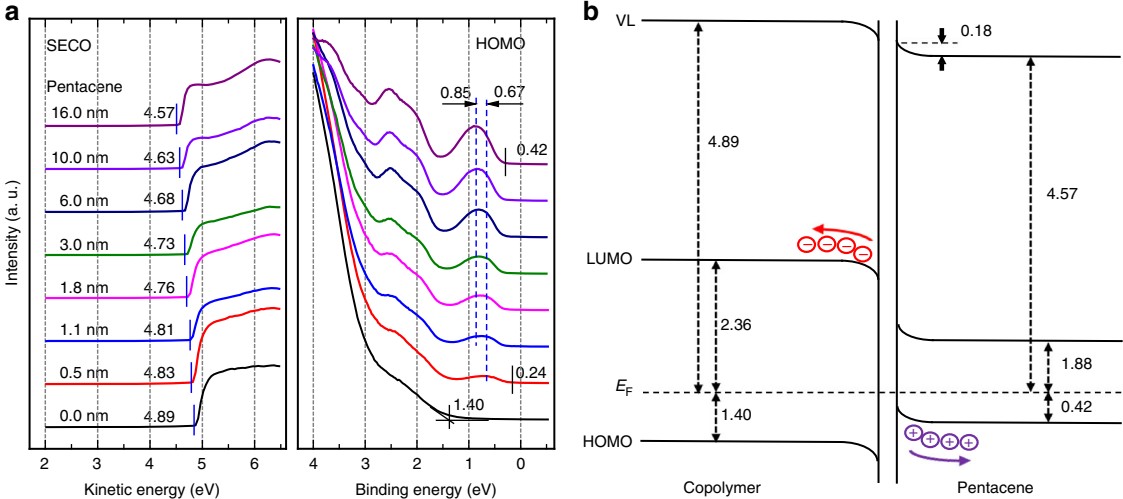

**Fig. 5** Interface characterization. **a** UPS spectra of incremental pentacene films on copolymer. **b** The derived energy-level diagram at the interface

smaller than PAA-based devices[38]. In addition, the devices showed outstanding operating stability in more than 4500 cycling tests of the transfer characteristics (Supplementary Fig. 13) and good environmental stability during shelf-life tests for 60 days (Supplementary Fig. 14, only 6% degradation of device performance was observed, compared to 13% degradation of PAA-based devices)[38].

**Interface electronic structures between copolymer and pentacene.** We carried out in-situ thickness-dependent ultraviolet

photoelectron spectroscopy (UPS) measurements to study the interface electronic structures between copolymer and pentacene. Figure 5a shows the UPS spectra presenting the evolution of secondary electron cut-off (SECO) region and highest occupied molecular orbital (HOMO) region at the pentacene/copolymer interface. It was obvious that the vacuum level (VL) decreased gradually from 4.89 to 4.57 eV after in-situ incremental deposition of pentacene on the copolymer, which indicated the charge (electron) transfer from pentacene to copolymer upon contact. Meanwhile, the HOMO of pentacene shifted 0.18 eV towards the

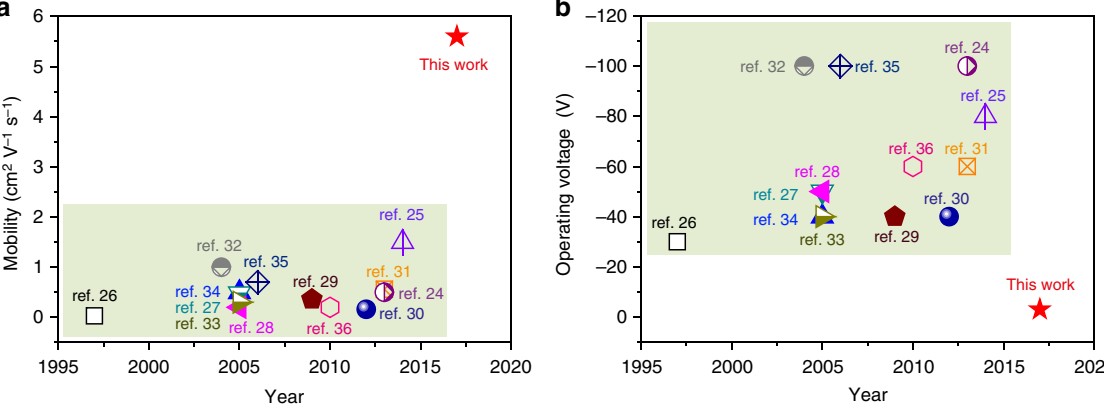

**Fig. 6** Comparison of device performances of this work and references. The distribution of mobility (**a**) and operating voltage (**b**) of OTFTs based on polyimide as dielectrics in the ref.[24-36] and this work

higher binding energy with its HOMO peak and leading edge from 0.67 and 0.24 eV to 0.85 and 0.42 eV below $E_F$. The derived schematic energy level diagram at pentacene/copolymer interface is depicted in Fig. 5b, where the HOMO positions are directly derived from the UPS measurements, and the lowest unoccupied molecular orbital (LUMO) edges are estimated by adding the optical band gaps of 3.46 and 2.30 eV for copolymer and pentacene, respectively, to their corresponding HOMO energy level. It can be concluded that when pentacene and copolymer come into contact, electrons would move from pentacene to copolymer across the interface and holes would be created (left) in pentacene. Consequently, the accumulation of holes and electrons at the interface leads to substantial band bending in both the pentacene and the copolymer layers. In a pentacene/copolymer OTFT, the free holes in pentacene side can be easily driven along the interface by an electric field applied across the source and drain electrodes. Hence, the pentacene/copolymer interface is favorable for charge transport process.

**Investigating the role of polar groups (−COOH/−CONH) on the surface**. We decreased the annealing temperature of PAA films to 180, 160, and 140 °C respectively for comparison. The capacitance of these copolymer layers were accordingly tested (from 20 Hz to 100 kHz) both in the air (under relative humidity of ~60%) and in the vacuum. Little difference of capacitance in the air and in the vacuum was observed (Supplementary Fig. 15 and Supplementary Fig. 16), which demonstrated outstanding robustness of these copolymer layers under different annealing treatments. ATR infrared spectroscopy (Supplementary Fig. 17a) showed that with the decrease of the annealing temperature (<200 °C), the imidization degree of the dielectric films dramatically reduced (46.44% for 180 °C, 41.43% for 160 °C, and 25.41% for 140 °C), which meant that the density of polar groups (−COOH/−CONH) on the surface correspondingly increased. With the same fabrication process, pentacene films exhibited higher crystallinity on all the dielectric layers (Supplementary Fig. 17b). From typical transfer curves (Supplementary Fig. 17c), it can be observed that the existence of polar groups (−COOH/−CONH) on the surface enhanced the carrier transport and the mobility of the devices ($C_{140°C}$, 30 nF cm$^{-2}$; $C_{160°C}$, 24 nF cm$^{-2}$; $C_{180°C}$, 22 nF cm$^{-2}$ from Supplementary Fig. 15) was inversely proportional to the imidization degree of the dielectric films (Supplementary Fig. 17d). Similar to PAA, the copolymer maintained surface polar groups (−COOH/−CONH) that could provide pronounced repulsive forces between the π-electron clouds of the pentacene backbone and the unshared electron pairs of oxygen atoms in the COOH-functionalized dielectric[38], leading

to more ordered packing and higher crystalline film with pentacene molecules standing on its surface. Therefore, the introduction of polar groups (−COOH/−CONH) plays a crucial role on the performance of this system.

**The effect of deposition rate of pentacene and further extended applications**. With increasing deposition rate, the grain size of pentacene on the surface of dielectric layer gradually decreased, which was observed on both copolymer surface (Supplementary Fig. 18a–c) and PI surface (Supplementary Fig. 18d–f). Simultaneously, the corresponding crystallinity of the pentacene films was also reduced with increasing deposition rate (Supplementary Fig. 19). Therefore, more grain boundaries and lower crystallinity of the pentacene films resulted in the decrease of the mobility (Supplementary Fig. 20). Even under this circumstance, the mobility value based on copolymer dielectrics was also higher than that with PI. In addition to pentacene, other organic semiconductors such as copper phthalocyanine (CuPc), copper hexadecafluorophthalocyanine ($F_{16}$CuPc) and 2, 6-diphenylanthracene (DPA) were also tested. As depicted in Supplementary Fig. 21, OTFTs with copolymer dielectric showed consistent improvement of device performance compared to that with PI dielectric, further confirming the general applicability of the copolymer. Moreover, this copolymer also allowed ambipolar operation and Supplementary Fig. 22 shows the typical transfer and output curve of this ambipolar transistor based on α, ω-Bis (biphenylyl) terthiophene (BP3T)[47]. Furthermore, the flexible OTFT devices based on pentacene were also characterized with a bending test (Fig. 4d) and the average mobilities decreased by less than 10% after bending the flexible devices for 1000 times over a 5 mm bending radius in parallel (red) and perpendicular (blue) directions (Fig. 4e), which verified the outstanding flexibility of these devices. All the above-mentioned results are significantly improved compared to the previously reported data based on PI dielectrics and pentacene (Fig. 6 and Supplementary Table 2).

**Fabrication and performance of flexible circuits**. Besides individual pentacene OTFTs, we also investigated the performance of logic circuits (invertors and oscillators) based on pentacene OTFTs on PET substrates. Figure 7a shows the flexible invertor arrays and the structure of a unipolar inverter. As shown in Fig. 7b, the inverters with the copolymer dielectrics on the PET substrate showed a switch response with a gain of ~15, implying its potential application in more complex logic circuits. The dynamic performance of the flexible low-voltage pentacene OTFTs was evaluated with a five-stage ring oscillator. Figure 7c shows a signal propagation delay of 100 μs at a supply voltage of

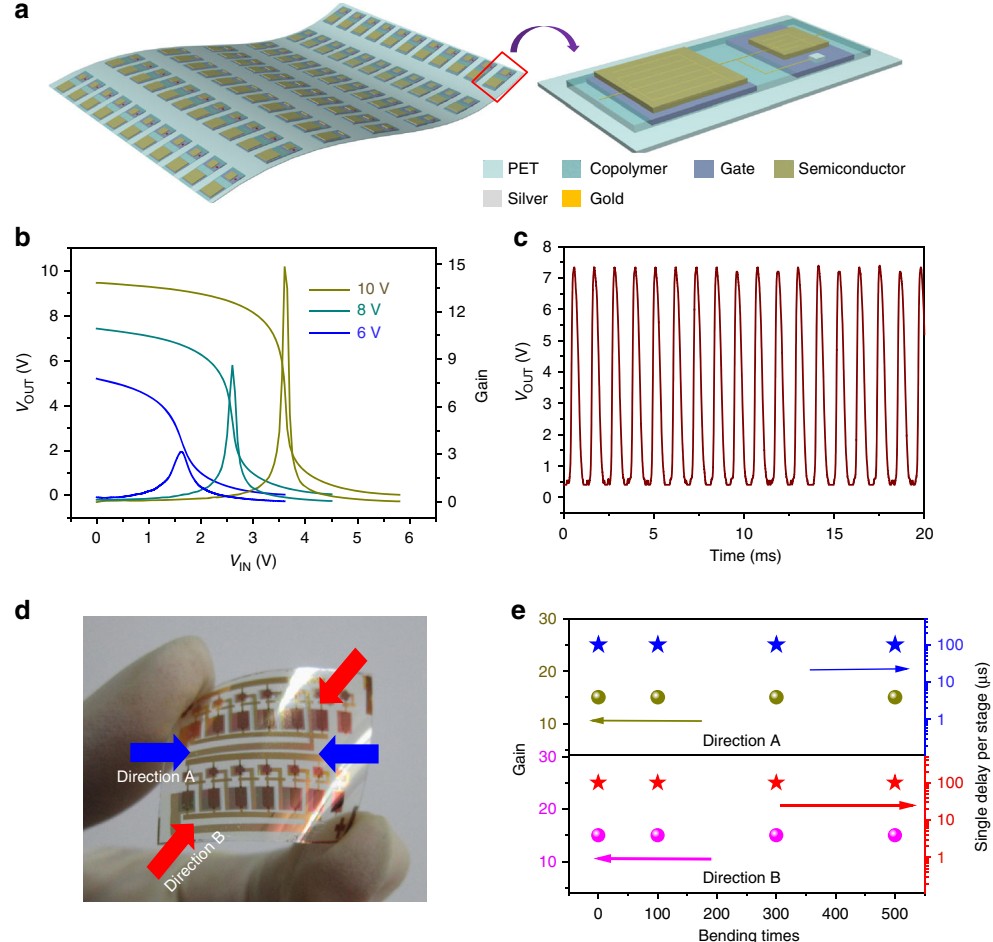

**Fig. 7** Performance of flexible circuits. **a** Flexible invertor arrays and the structure of a unipolar inverter. **b** Output voltage and signal gain as a function of input voltage with supply voltage of 6, 8, and 10 V. **c** Five-stage ring oscillators and their representative electrical characteristic based on pentacene. **d** The bending test (direction A and direction B) of the circuits. **e** Plots of gain and single delay per stage versus bending times

10 V. To test the flexibility of circuits, we bend the devices over a 5 mm bending radius by using a cylindrical object at different directions (Fig. 7d, direction A and direction B). As shown in Fig. 7e, there was negligible change for both the gain and the single delay per stage of the circuits after bending for 500 times. These bending tests further confirmed the outstanding flexibility of these circuits, indicating their promising application in more complex integrated circuits.

## Discussion

In summary, in this study we introduce a simple and highly efficient strategy to prepare a new structured copolymer for robust dielectric layers, which not only contains the phenyl rings and alicyclic rings for high chain packing density that improves the insulating property, but also bears the strong polar groups (−COOH/−CONH) favoring molecular packing and charge transport on its surface. A good balance of these two properties achieved by the copolymer dielectric contributes to flexible OTFTs with unprecedented performance (mobility up to 5.6 cm$^2$ V$^{-1}$ s$^{-1}$ with operation voltage as low as 3 V for pentacene) and outstanding stability. Furthermore, the general applicability of the copolymer dielectric is verified and flexible logic circuits such as inverters and oscillators are successfully manufactured. The experimental results indicate exciting prospects of the copolymer for plastic high-performance integrated electronics.

## Methods

**XPS and UPS measurements**. The samples were prepared by spin-coating 5~7 nm PAA film on the surface of clean ITO glass substrates, which were then annealed to form the copolymer films. Then pentacene layers with incremental thickness were in-situ deposited in the evaporation chamer connected to the photoelectron spectroscopy. X-ray photoelectron spectroscopy (XPS) was performed in a Kratos AXIS ULTRA DLD ultrahigh vacuum surface analysis system at the base pressure of $2 \times 10^{-9}$ Torr with a monochromatic Al K$\alpha$ (1486.6 eV) as the excitation source. Ultraviolet photoelectron spectroscopy (UPS) measurements were carried out in a Kratos AXIS ULTRA DLD photoelectron spectroscopy at $3 \times 10^{-8}$ Torr with a He-discharge lamp (21.22 eV) as the excitation source with sample bias voltage of -9 V.

**Capacitance measurement**. The frequency-dependent capacitance was measured with a HP 4284 A Precision LCR Meter in a frequency range of 20 Hz to 100 kHz.

**ATR measurement**. Attenuated total reflection (ATR) infrared spectroscopy was measured by Fourier Transform Infrared Spectroscopy (FTIR; TENSOR-27, BRUKER). The samples were prepared on the glass wafer and then treated under different annealing temperatures.

**Fabrication and characterizations of pentacene-based OTFTs**. Bottom-gate top-contact pentacene thin film transistors were fabricated by the following procedures: (1) ITO/PET substrates used in the study were successively cleaned with pure water, acetone, pure ethanol, and pure isopropanol and then dried with nitrogen. The surface of ITO/PET substrate was treated with O$_2$ plasma (50 W, 1 min). Here plasma treatment was carried out using Gala Instrument Prep2; (2) PAA solution was synthesized in the lab and spin-coated onto the surface of ITO/PET to form 160 nm film and then this PAA film was annealed in the air to be in-situ imidized into random copolymer film; (3) the substrate was transferred to a vacuum chamber and 50 nm pentacene was deposited with the deposition rate of 0.05 Å s$^{-1}$; (4) 20 nm thickness Au was deposited on the pentacene surface using

copper mask (with the deposition rate of 0.1 Å s$^{-1}$) to finish the device. The morphology of dielectrics and pentacene was characterized by Atomic force microscopy (AFM) using a Nanoscopy IIIa instrument (USA). X-ray diffraction (XRD) of the pentacene layers on the dielectrics was recorded on a D/max2500 with a Cu Kα source ($k = 1.541$ Å). The electrical characteristics of the OTFT devices were measured at room temperature in air and in the vacuum by using a Keithley 4200 SCS semiconductor parameter analyzer and a Micromanipulator 6150 probe station. The mobility was extracted from the saturation region by using the equation of $I_{DS} = (W/2 L) C_i \mu (V_G - V_T)^2$.

**Fabrication and measurement of pentacene-based circuits**. Pentacene-based circuits were fabricated with the following steps: (1) the patterned gate electrodes (Al, Ag, or Au) for invertors and oscillators were deposited (50 nm) through a metal mask on the PET substrate; (2) PAA solution was synthesized in the lab and spin-coated onto the surface of ITO/PET and then this PAA film was annealed in the air to be in-situ imidized into random copolymer film; (3) O$_2$ plasma was used to remove the dielectric layer above the connecting parts by means of a metal mask to protect the rest of the circuit; (4) 100 nm Ag was deposited into the connecting parts; (5) 50 nm pentacene was patterned with a metal mask on the dielectrics; (6) the last layer (source/drain electrodes) and the rest interconnecting lines were finished by depositing Au. The electrical characteristics of the invertors were recorded at room temperature in air by four probes using a Keithley 4200 SCS semiconductor parameter analyzer and a Micromanipulator 6150 probe station. The electrical characteristics of the oscillators were measured at room temperature in air by oscilloscope DPO 2012.

**Data availability**. The authors declare that the data supporting the findings of this study are available from the corresponding author upon reasonable request.

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

## Acknowledgements

The GIXRD data were obtained at 1W1A, Beijing Synchrotron Radiation Facility. The authors gratefully acknowledge the assistance of scientists of Diffuse X-ray Scattering Station during the experiments. The authors acknowledge financial support from the Deutsche Forschungsgemeinschaft (SFB 858 and TRR 61), the Ministry of Science and Technology of China (Grants 2016YFB0401100, 2017YFA0204503, 2017YFA0207500), National Natural Science Foundation of China (51725304, 51673114, 51633006), the Strategic Priority Research Program (Grant No. XDB12030300). The authors are grateful to Prof. Xingyi Huang (Shanghai Jiao Tong University) and Dr. Zupan Mao (Institute of Chemistry, Chinese Academy of Sciences) for profound discussions and Ms. Qin Tang (Nanjing University of Science& Technology) for figures design.

## Author contributions

H.F., H.W., L.T. and J.D. conceived and designed the experiments. J.D., K.Z., J.L., Z.Z., Z.X., L.L. and H.D. performed the experiments. Z.Y. contributed to UPS and XPS experimental data and analysis. T.G. and W.D. synthesize the precursor poly (amic acid) (PAA). C.M. and M.Q. contributed to the measurements of the capacitance. J.D., L.T. and Z.Y. discussed the results and co-wrote the manuscript.

## Additional information

**Competing interests:** The authors declare no competing interests.

