## [Peer Review File · Nature Communications]

Reviewer #1 (Remarks to the Author):

Comments

The manuscript entitled “Copolymer Dielectrics with Balanced Chain-Packing Density and Surface Polarity for High-Performance Flexible Organic Electronics” presents the synthesis of a novel dielectric copolymer by introducing partial imidization to polyamic acid (PAA) and the evaluation of its performance in OFET devices as the gate insulator. The authors compared the partially imidized copolymer to its totally imidized counterparts (PI) in their respective OFET devices. It was found that the copolymer exhibited an insulating property between PAA and PI. Furthermore, they discovered that the morphology of pentacene coated on PI and the copolymer substrate showed drastic difference, i.e. more edge-on and more crystalline on the copolymer substrate. A relatively high mobility ($5.6 \text{ cm}^2 \text{ V}^{-1} \text{ s}^{-1}$) at a relatively low operating voltage (3V) was demonstrated for the OFET using the copolymer as dielectric layer, much higher than the OFET using PI as the gate dielectric. Overall, the paper is interesting and should have important impacts to the OFET field. The paper could be considered for publication if the following issues can be addressed.

1. The authors excluded PAA in their OFET device performance comparison. The motivation of partial imidization was to, quoting the authors, “balance the chain-packing density guaranteeing insulating properties and surface polarity optimizing molecular packing of organic semiconductors”. Whereas, the authors only showed that the insulating property of the copolymer was better than PAA, but PAA has more polar groups than the copolymer, so the morphology of pentacene grown on the PAA underlayer could be even better than the copolymer, which could easily overwhelm the benefit provided by the high insulating property from the copolymer. In their previous publication (J. Am. Chem. Soc. 2017, 139, 2734–2740), the pentacene-based OFET using PAA as the substrate was demonstrated to be as high as $30.6 \text{ cm}^2 \text{ V}^{-1} \text{ s}^{-1}$ at the same operating voltage used in this manuscript. But the authors chose to compare the copolymer only to the relatively poor-performing PI instead of the high-performance PAA.
2. The current density of a metal/dielectric device depends on the thickness of the dielectric layer. The authors should provide the actual thicknesses of PAA, PI and the copolymer with error bars for Figure 2a, and the technique used for thickness measurement.
3. Figure S15 should include error bars for each mobility.
4. The authors calculated the surface energies of PI and the copolymer, which are fine. But they used the surface energies of different pentacenes grown on different gate insulators to explain the difference in the arrangement of pentacene molecules. This argument is wrong. The high surface energy of pentacene on PI is the consequence of the morphology of pentacene formed on PI, not the cause of it. Similarly, the relatively lower surface energy of pentacene on the copolymer ($\sim 31 \text{ mJ/m}^2$) is the result of the morphology of pentacene, not the cause. If the authors want to explain the difference in the growth mode of pentacene on different gate insulators, they should analyze the interfacial free energy or the work of adhesion. Consider γ_p , and γ_g as the surface energies of pentacene and gate insulator, and γ_i as the interfacial free energy. The Frank-van der Merwe mode, or the layer-by-layer growth mode should occur when $\gamma_p + \gamma_i \leq \gamma_g$ while the Volmer-Weber growth

mode, or the 3D island mode should occur when $\gamma_p + \gamma_i > \gamma_g$. To explain the difference in the growth mode, one should analyze the interfacial energy or the work of adhesion, instead of measuring the contact angle of a pentacene grown onto a specific substrate.

Reviewer #2 (Remarks to the Author):

In this communication, the authors developed a dielectric material based on poly(amic acid) (PAA) and polyimide (PI). The partially imidized PAA/PI copolymer showed not only good insulating property but compatibility with organic semiconductors. The pentacene-based transistor device using this PAA/PI copolymer dielectric layer exhibited a mobility of $5 \text{ cm}^2\text{V}^{-1}\text{s}^{-1}$ with an operating voltage of 3 V. In addition, the device can be integrated into flexible electronics and possessed stable electrical characteristics.

Although the flexible devices with PAA/PI copolymer dielectric showed good performance, it still unclear why this PAA/PI low-k dielectric materials can achieve high-performance transistor with low operating voltage. I suspect that the dielectric layer has double-layer capacitor effect, which needs to be carefully studied as suggested below. A general mechanism between chemical structure and electrical property is needed. Also, the novelty of the PAA/PI material design is little since similar PI polymers were well-characterized and were already applied to the flexible transistor (e.g. *Macromol. Rapid. Commun.*, 2014, 35, 1039; *PANS*, 2004, 101, 9966). The manuscript thus is unsuitable for the publication in *Nature Communications*.

Several details related to materials and devices should be still elucidated:

1. The authors have only one copolymer that is PAA:PI = 11%:89%. What happens with PAA/PI copolymer with different PAA ratios? This should be studied systematically. A deeper understanding on how the PAA groups help to improve the dielectric properties is needed.
2. The PAA/PI material that was used in this study is a well-known PI system, and has been classified as a low-k material (*Prog. Polym. Sci.*, 2001, 26, 3-65). Similarly, the dielectric constant of the PAA/PI

copolymer in this study is close to that of SiO₂. Such low-k dielectric layers usually need high operation voltage to drive the transistor devices. I suspect the polar groups in the PAA/PI copolymer layer can easily trap moisture or other impurities in air during the measurement, and further change the dielectric as well as transistor performance. The transistor and capacitance measurement in nitrogen atmosphere or under vacuum should be performed, and the authors should carefully explain why this low-k dielectric material can lead to low-voltage-operated transistor device. Furthermore, the authors should measure capacitance as a function of dielectric thickness. This can provide information on whether double-layer capacitor effect is in play in this dielectric. In that case, the capacitance reported may be underestimated and the mobility may be grossly overestimated. The lack of double-layer capacitor effect needs to be confirmed before presenting any mobility values.

3. In Figure 3a, the polymer structure should be PAA/PI copolymer, not the PAA polymer.

Reviewer #3 (Remarks to the Author):

In the manuscript a copolymer based dielectric is presented, combining high packing density and polar groups. As a result the dielectric exhibits low leakage currents and enhances the packing of organic semiconducting molecules leading to high carrier mobilities ($\sim 5 \text{ cm}^2/\text{Vs}$). The paper contains a large number of experiments and the OFETs based on this copolymer dielectric show very good properties. I have only a few remarks:

-from the given capacitance values I estimate that the relative dielectric constant is around 4, is that correct?

-On this dielectric, the mobility of pentacene is about 10 times enhanced as compared to other PI dielectrics. As a result, due to the high mobility an identical source-drain current can be obtained at lower gate bias. This is the reason why the OFETs here have an operating voltage of 3V. Similar low gate bias operation can be achieved with other dielectric/semiconductor combinations that give a high mobility. What is more interesting is that with a high mobility also higher currents can be induced, for example to drive an OLED. What is then important to know is how large the break down field (voltage over thickness) is, such that one can estimate what the maximal current is at which

such an OFET can operate. The break down field is not mentioned and also not compared to other dielectrics. This should be added.

-Gate insulators often trap electrons due to for example OH groups, leading to p-type operation only. Other dielectrics also allow electron transport, leading to ambipolar transistors. Does the dielectric presented here also allow ambipolar operation?

With these comments answered the paper can be accepted for publication.

Judgement: Accept with minor corrections

For Referee #1

Dear Referee,

We greatly appreciated your encouraging and valuable suggestions for our manuscript (NCOMMS-17-23269). According to your comments, we carried out more experiments and made corresponding revisions marked red in the manuscript. Thanks again for your insightful comments!

Comment 1: The authors excluded PAA in their OFET device performance comparison. The motivation of partial imidization was to, quoting the authors, “balance the chain-packing density guaranteeing insulating properties and surface polarity optimizing molecular packing of organic semiconductors”. Whereas, the authors only showed that the insulating property of the copolymer was better than PAA, but PAA has more polar groups than the copolymer, so the morphology of pentacene grown on the PAA under layer could be even better than the copolymer, which could easily overwhelm the benefit provided by the high insulating property from the copolymer. In their previous publication (J. Am. Chem. Soc. 2017, 139, 2734–2740), the pentacene-based OFET using PAA as the substrate was demonstrated to be as high as $30.6 \text{ cm}^2 \text{ V}^{-1} \text{ s}^{-1}$ at the same operating voltage used in this manuscript. But the authors chose to compare the copolymer only to the relatively poor-performing PI instead of the high-performance PAA.

Our reply: Thank you very much for your professional comments! It is indeed important to make detailed comparison between the copolymer and PAA. Besides the insulating property shown in the Figure 2a, we also compared the air stability of the copolymer and PAA (Supplementary Fig. 15). It was found that ~13% degradation of device performance based on PAA dielectrics was observed after 60 days. And only ~10% degradation of device performance based on copolymer insulating layers was observed after 100 days. This result indicated that the copolymer design could improve the stability of the devices. What’s more, in our revised manuscript, we did more experiments to compare these two dielectric layers, such as the break-down field (Figure 2b) and hysteresis-effect (Supplementary Fig. 14) measurements. From the break-down field measurement, it clearly showed that PAA could only withstand less than 400 mV cm^{-1} electric field, while the copolymer film with same thickness exhibited much higher break-down field (about 650 mV cm^{-1}) which is almost twice that of PAA and very close to fully cross-linked PI system (about 660 mV cm^{-1}). In addition, the hysteresis-effect measurements also proved that the copolymer dielectrics could improve the operation stability of the device. Collectively, the above-mentioned experimental results indicated that except the mobility value (lower than the PAA-based devices), the copolymer with balanced insulating property and surface polarity showed many superior performances (the insulating property, air stability, break-down field and hysteresis-effect) than that with PAA dielectrics.

Our revision: We have added the break-down field (Figure 2b) and hysteresis-effect (Supplementary Fig. 14) data in our revised manuscript. In page 6, we added the following revised contents. “Besides, PAA dielectric layer could only withstand less than 400 mV cm^{-1} electric field, while this copolymer

film with same thickness exhibited much higher break-down field (about 650 mV cm^{-1}), which is almost twice that of PAA and very close to fully cross-linked PI system (about 660 mV cm^{-1}) (Figure 2b).” In page 10, “Compared with the devices based on PAA, this type of copolymer-based transistor showed negligible hysteresis-effect (Supplementary Fig. 14) and higher air stability (Supplementary Fig. 15).”

New Figures:

Figure 2. Current density, Break-down field, XRD, GIXRD and AFM characterizations. a) Current density of the dielectric layers at the bias voltage of 5V. Inset, an Au/Dielectrics (160 nm)/ITO/PET sandwiched device structure for test. (b) Current density as a function of electric field, (c) XRD patterns of pentacene films (50 nm) grown on copolymer and PI surfaces, (d) 2D GIXRD patterns of pentacene films (50 nm) on the surface of copolymer and PI. (e)- (j) AFM images of pentacene films grown on dielectric substrates with different thicknesses (4.5 nm, 15 nm and 50 nm): (e, g, i) copolymer; (f, h, j) PI.

Figure S14. The hysteresis-effect characterization of pentacene OTFTs based on (a) PAA and (b) copolymer dielectrics.

Comment 2: The current density of a metal/dielectric device depends on the thickness of the dielectric layer. The authors should provide the actual thicknesses of PAA, PI and the copolymer with error bars for Figure 2a, and the technique used for thickness measurement.

Our reply: Thank you for this suggestion. The thickness of the dielectric layer is important for the comparison of the current density among the different dielectric materials. We have used atomic force microscopy (AFM) to measure the thickness of the dielectric layers with error bars, which have been added in Supplementary Fig. 3.

Our revision: In page 6, we added the revised contents. “To investigate the dielectric properties of these insulator films, current density and break-down field were tested. A sandwiched device structure of Au/insulator films/indium tin oxide (ITO)/PET was adopted (Fig. 2a inset) and the thickness of each insulator film (around 160 nm) was measured by atomic force microscopy (AFM) (Supplementary Fig. 3).”

New Figures:

Figure S3. The AFM images of dielectric films with the thickness around 160 nm.

Comment 3: Figure S15 should include error bars for each mobility.

Our revision: Yes, we should have done this. Thank you very much! We have added the error bars for each mobility value.

New Figures:

Figure S20. Mobilities based on different organic semiconductors with copolymer and PI dielectrics, respectively

Comment 4: The authors calculated the surface energies of PI and the copolymer, which are fine. But they used the surface energies of different pentacene grown on different gate insulators to explain the difference in the arrangement of pentacene molecules. This argument is wrong. The high surface energy of pentacene on PI is the consequence of the morphology of pentacene formed on PI, not the cause of it. Similarly, the relatively lower surface energy of pentacene on the copolymer ($\sim 31 \text{ mJ/m}^2$) is the result of the morphology of pentacene, not the cause. If the authors want to explain the difference in the growth mode of pentacene on different gate insulators, they should analyze the interfacial free energy or the work of adhesion. Consider γ_p , and γ_g as the surface energies of pentacene and gate insulator, and γ_i as the interfacial free energy. The Frank-van der Merwe mode, or the layer-by-layer growth mode should occur when $\gamma_p + \gamma_i \leq \gamma_g$ while the Volmer-Weber growth mode, or the 3D island mode should occur when $\gamma_p + \gamma_i > \gamma_g$. To explain the difference in the growth mode, one should analyze the interfacial energy or the work of adhesion, instead of measuring the contact angle of a pentacene grown onto a specific substrate.

Our reply: Thank you for your insightful suggestions! Yes, you are right! We should use the interfacial free energy or the work of adhesion to explain the difference in the growth mode of pentacene. What you have mentioned in the comments is very helpful. In our manuscript, what we wanted to show was not the explanation of the difference in the arrangement of pentacene molecules, but that the matching surface energy between the insulator and pentacene is very important for more efficient transistor channels. We feel sorry for this wrong description which may have caused your misunderstanding. We have corrected this description. Thank you very much!

Our revision: In page 8, we added the revised contents. “Compared with a large difference in surface energy between PI dielectric (29.4 mJ cm^{-2}) and overlying pentacene (50.5 mJ cm^{-2}), the copolymer had a surface energy (28.5 mJ cm^{-2}) that was similar to pentacene layer (31.4 mJ cm^{-2}). The matching of surface energy between the insulator and pentacene is very important due to possibly more efficient transistor channels developed from this interface.^{14, 44}”

Thanks again for your great contribution to our manuscript!

Sincerely yours,

Wenping Hu

For Referee #2

Dear Referee,

We greatly appreciate your insightful comments and suggestions for our manuscript (NCOMMS-17-23269). In accordance with your suggestions, we have carried out more detailed investigations and obtained more positive supporting results, which help us further improve the quality of our manuscript. Now we feel confident to have addressed all the issues and our point-to-point replies and revisions (marked in red) are shown as follows. Thank you very much again for your insightful comments and great contribution to our manuscript!

Comment 1: In this communication, the authors developed a dielectric material based on poly (amic acid) (PAA) and polyimide (PI). The partially imidized PAA/PI copolymer showed not only good insulating property but compatibility with organic semiconductors. The pentacene-based transistor device using this PAA/PI copolymer dielectric layer exhibited a mobility of $5 \text{ cm}^2\text{V}^{-1}\text{s}^{-1}$ with an operating voltage of 3 V. In addition, the device can be integrated into flexible electronics and possessed stable electrical characteristics. Although the flexible devices with PAA/PI copolymer dielectric showed good performance, it still unclear why this PAA/PI low-k dielectric materials can achieve high-performance transistor with low operating voltage. I suspect that the dielectric layer has double-layer capacitor effect, which needs to be carefully studied as suggested below. A general mechanism between chemical structure and electrical property is needed. Also, the novelty of the PAA/PI material design is little since similar PI polymers were well-characterized and were already applied to the flexible transistor (e.g. *Macromol. Rapid. Commun.*, 2014, 35, 1039; *PANS*, 2004, 101, 9966). The manuscript thus is unsuitable for the publication in *Nature Communications*.

Our reply: Thank you for your valuable comments. The major novelty of our work is to offer a simple and highly efficient strategy for the preparation of a new structured copolymer dielectric for flexible organic thin-film transistors (OTFTs) applications. By changing the annealing temperature, the imidization degree of the polymers can be precisely controlled. More interestingly, the two monomers produced with this method play different roles: one could increase the chain-packing density guaranteeing sufficient insulating properties and the other is responsible for surface polarity optimizing molecular packing of organic semiconductors. Benefitting from a good balance of the above two aspects, we are happy to see much higher performances of the OTFT devices with this copolymer dielectric than the previously reported results based on polyimide dielectrics shown in Figure 4 and Table S2, not only the mobility, but also the operation voltage.

Although PI polymers were well-characterized and had already been applied to the flexible transistor (e.g. *Macromol. Rapid. Commun.*, 2014, 35, 1039; *PNAS*, 2004, 101, 9966), the reported devices (either served as the buffer layer or the independent dielectric layer) were operated at high voltage (above 40 V),

resulting in high energy consumption mainly due to the use of thick PI films (above 500 nm) for preventing leakage current. Another issue is that the previously reported structure of polyimide dielectric generally resulted in low-performance devices (mobilities not exceeding $1.0 \text{ cm}^2 \text{ V}^{-1} \text{ s}^{-1}$) probably due to the mismatch of surface energy between the dielectric and the overlying organic semiconductor. Against this background, we believe our design of this copolymer for substantial improvement of OTFT performances makes a new breakthrough for polyimide-based devices and would promote the development of polyimide-based flexible devices.

As we know, the capacitance of plate capacitor can be calculated as $C = \epsilon_0 (\kappa A/d)$. In order to increase the capacitance for low operating voltage, there are mainly two approaches: one is to use dielectric materials with a relatively high dielectric constant (κ); the other is to reduce the thickness of the dielectric layer, which is especially helpful for most of the low- κ materials. In our work, we reduced the thickness of the dielectric layer to about 160 nm for low-voltage operation. In our revised manuscript, we further investigated different imidization degree of the copolymer and found that the polar groups ($-\text{COOH}/-\text{CONH}$) played an important role on the performance of the device. In addition, the capacity values of copolymer dielectrics as a function of thickness in air and in the vacuum were tested, excluding the possibility of double-layer capacitor effect. Experimental details are given below.

Comment 2: The authors have only one copolymer that is PAA:PI = 11%:89%. What happens with PAA/PI copolymer with different PAA ratios? This should be studied systematically. A deeper understanding on how the PAA groups help to improve the dielectric properties is needed.

Our reply: Thank you very much for your professional suggestions! Different imidization degree can be easily achieved by controlling the annealing temperature. We carried out more experiments based on your comments. For a deep understanding of the effect of polar groups ($-\text{COOH}/-\text{CONH}$), another three annealing temperatures (180 °C, 160 °C and 140 °C) were adopted. We utilized attenuated total reflection infrared spectroscopy for characterization (Supplementary Fig. 16a), and the imidization degree for these three new dielectric layers were calculated to be about 46.44% for 180 °C, 41.43% for 160 °C and 25.41% for 140 °C, respectively. Pentacene films (50 nm in thickness) exhibited high crystallinity on all of these dielectric layers confirmed by XRD (Supplementary Fig. 16b). It was observed that the existence of polar groups ($-\text{COOH}/-\text{CONH}$) on the surface enhanced the carrier transport and the mobility of the devices was inversely proportional to the imidization degree of the dielectric films (Supplementary Fig. 16c and d). Similar to PAA, We believe that this copolymer maintained polar groups ($-\text{COOH}/-\text{CONH}$) on the surface that could provide pronounced repulsive forces between the π -electron clouds of pentacene backbone and the unshared electron pairs of oxygen atoms in the COOH-functionalized dielectric, leading to more ordered packing and higher crystalline film with pentacene molecules standing on its surface.

Our revision: In page 10 and page 11, we added the revised contents. “To further investigate the role of polar groups (–COOH/–CONH) on the surface, we decreased the annealing temperature of PAA films to 180 °C, 160 °C and 140 °C respectively for comparison. ATR infrared spectroscopy (Supplementary Fig. 16a) showed that with the decrease of the annealing temperature (< 200 °C), the imidization degree of the dielectric films dramatically reduced (46.44% for 180 °C, 41.43% for 160 °C and 25.41% for 140 °C), which meant that the density of polar groups (–COOH/–CONH) on the surface correspondingly increased. With the same fabrication process, pentacene films exhibited higher crystallinity on all the dielectric layers (Supplementary Fig. 16b). From typical transfer curves (Supplementary Fig. 16c), it can be observed that the existence of polar groups (–COOH/–CONH) on the surface enhanced the carrier transport and the mobility of the devices was inversely proportional to the imidization degree of the dielectric films (Supplementary Fig. 16d). Similar to PAA, the copolymer maintained surface polar groups (–COOH/–CONH) that could provide pronounced repulsive forces between the π -electron clouds of pentacene backbone and the unshared electron pairs of oxygen atoms in the COOH-functionalized dielectric,³⁸ leading to more ordered packing and higher crystalline film with pentacene molecules standing on its surface. Therefore the introduction of polar groups (–COOH/–CONH) plays a crucial role on the performance of this system.”

New figure:

Figure S16 | The characterization of PAA films under different annealing temperature. (a) Attenuated total reflection (ATR) infrared spectroscopy of PAA with different imidization temperature.

(b) XRD patterns of pentacene films (50 nm) grown on these three dielectrics. (c) Typical transfer curves of the OTFTs with 50 nm pentacene and a channel dimension of $W = 240 \mu\text{m}$, $L = 30 \mu\text{m}$. (d) Mobility as a function of annealing temperature.

Comment 2: The PAA/PI material that was used in this study is a well-known PI system, and has been classified as a low-k material (Prog. Polym. Sci., 2001, 26, 3-65). Similarly, the dielectric constant of the PAA/PI copolymer in this study is close to that of SiO_2 . Such low-k dielectric layers usually need high operation voltage to drive the transistor devices. I suspect the polar groups in the PAA/PI copolymer layer can easily trap moisture or other impurities in air during the measurement, and further change the dielectric as well as transistor performance. The transistor and capacitance measurement in nitrogen atmosphere or under vacuum should be performed, and the authors should carefully explain why this low-k dielectric material can lead to low-voltage-operated transistor device. Furthermore, the authors should measure capacitance as a function of dielectric thickness. This can provide information on whether double-layer capacitor effect is in play in this dielectric. In that case, the capacitance reported may be underestimated and the mobility may be grossly overestimated. The lack of double-layer capacitor effect needs to be confirmed before presenting any mobility values.

Our reply: Thank you for your insightful comments! An effective way to obtain low operating voltage is to increase the dielectric capacitance, which (for a plate capacitor) can be calculated as $C = \epsilon_0 \kappa A/d$. Accordingly, there are two approaches for preparing gate insulators to achieve low operating voltage: one is to use dielectric materials with a relatively high dielectric constant (κ); the other is to reduce the thickness of the dielectric layer, which is especially helpful for most of the low-k materials. Take the widely used low-k SiO_2 ($k=3.9$) as an example, low-voltage operation can be realized by reducing the SiO_2 thickness. (Adv. Mater. 2012, 24, 2159–2164; Adv. Mater. 2012, 24, 3053–3058). What's more, there are a number of papers reporting low-k polymer dielectric materials for low-voltage operation by decreasing their thickness and the operation voltage could be as low as 1 V. (J. Am. Chem. Soc. 2005, 127, 10388–10395; Appl. Phys. Lett. 2006, 88, 242113; Appl. Phys. Lett. 2006, 89, 183516; Org. Electron., 2009, 10, 174–180; Chem. Commun., 2010, 46, 3961–3963; Chem. Mater. 2010, 22, 1559–1566; J. Mater. Chem., 2010, 20, 9047–9051; ACS Appl. Mater. Interfaces 2012, 4, 3261–3269; Appl.

Phys. Lett. 2012, 101, 033303). Therefore, the use of low-k dielectric materials and the realization of low-voltage operation are not contradictory.

In previous reports, polyimide (PI) was usually used as the buffer layer (Prog. Polym. Sci., 2001, 26, 3-65) with high operation voltage. The reported devices based on independent PI dielectric layer mostly chose the thickness above 500 nm (Chem. Mater. 1997, 9, 1299; Adv. Funct. Mater. 2005, 15, 619; Appl. Phys. Lett. 2005, 86, 133508; Org. Electro. 2009, 10, 12; Org. Electro. 2012, 13, 1665; Phys. Chem. Chem. Phys. 2013, 15, 950; Adv. Funct. Mater. 2014, 24, 3783) for preventing leakage current and the corresponding operation voltages were above 40 V. In addition, the reported structure of polyimide generally resulted in low-performance devices (mobilities not exceeding $1.0 \text{ cm}^2 \text{ V}^{-1} \text{ s}^{-1}$) probably due to the mismatch of surface energy between PI and organic semiconductors. Under these circumstances, we combined the advantages of the PI and PAA for the design of a new kind of copolymer with balanced chain-packing density and surface polarity, at the same time reducing the thickness of this copolymer to about 160 nm, thus successfully achieving low-voltage operation and higher mobility. We believe such a design concept can significantly push forward the development of the polyimide-based insulators and can be potentially applied to other systems.

There are polar groups in copolymer layer and in order to verify whether this copolymer could easily trap moisture or other impurities in air during the measurement, the transistor and capacitance measurements in the vacuum were performed. As shown in Supplementary Fig. 8 and Supplementary Fig. 9, the device performance showed no obvious difference between the conditions in the air and in the vacuum (only that the threshold voltage changed from -0.59V to -0.39V). Furthermore, the air-stability test (Supplementary Fig.13) showed only ~10% degradation of device performance after stored in the air for more than 100 days. This also indicated that our copolymer would not easily trap moisture or other impurities.

In order to verify whether there exists double-layer capacitor effect in the copolymer, we did the capacitance measurements both in the air and in the vacuum. Five different thicknesses (250 nm, 350nm, 650 nm, 900 and 1100 nm) of the copolymer films were tested. Supplementary Fig. 8 shows capacitance values as a function of the thickness of dielectrics. It was clear that there was little difference between

that in the air and in the vacuum (especially in the low-frequency region), thus excluding the possibility of double-layer capacitor effect in this system.

Our revision: In page 9, we added the revised contents. “To more accurately calculate the dielectric constant, capacitance values of the copolymers with various thicknesses were tested both in the air and in the vacuum (Supplementary Fig. 8). It is clear that negligible difference of capacitance was observed, especially in the low frequency region. As a result, the dielectric constant of this copolymer can be calculated around 4.” “What’s more, the devices were stable in the atmosphere environment with little influence by the moisture or other impurities in air, as indicated by their almost the same performance in the air and in the vacuum (Supplementary Fig. 9).”

New figures:

Figure S8 | The frequency dependence of capacitance for copolymer dielectrics with different thickness under 200 °C annealing temperature. Negligible difference of capacitance was observed, especially in the low frequency region.

Figure S9. The electrical characterization of pentacene OTFTs based on copolymer dielectric in the air and in the vacuum.

Figure S13 |The air stability characterization of pentacene OTFTs based on copolymer dielectric. Mobility and ON/OFF ratio dependence on time.

Comment 3: In Figure 3a, the polymer structure should be PAA/PI copolymer, not the PAA polymer

Our reply: Thank you for pointing out this mistake. We are sorry for our negligence and have revised this figure.

New Figure:

Figure 3 | Devices structure and performance. (a) The preparation process for large-area flexible OTFT arrays, (b) Distribution of device mobility, (c) Typical transfer curve of the OTFT with 50 nm pentacene and a channel dimension of $W = 240 \mu\text{m}$, $L = 30 \mu\text{m}$. The gate current as a function of gate-source voltage is shown in purple, (d) A photograph of flexible devices for investigating the effect of bending times, (e) Plots of mobility versus bending times on PET substrate based on copolymer insulating layers

Thanks again for your great contribution to our manuscript!

Sincerely yours,

Wenping Hu

For Referee #3

Dear Referee,

We greatly appreciate your encouraging and valuable suggestions, which help us improve the quality of our manuscript. According to your suggestions we have revised our manuscript.

Comment 1: from the given capacitance values I estimate that the relative dielectric constant is around 4, is that correct?

Our reply: Yes, that's correct! In order to more accurately calculate the dielectric constant, capacitance values of the copolymers with various thicknesses were tested both in the air and in the vacuum (Supplementary Fig. 8). It is clear that negligible difference of capacitance was observed, especially in the low frequency region. According to the equation $C = \epsilon_0 (\kappa A/d)$, the dielectric constant of this copolymer can be calculated around 4.

New figure:

Figure S8 | The frequency dependence of capacitance for copolymer dielectrics with different thickness under 200 °C annealing temperature. Negligible difference of capacitance was observed, especially in the low frequency region.

Comment 2: On this dielectric, the mobility of pentacene is about 10 times enhanced as compared to other PI dielectrics. As a result, due to the high mobility an identical source-drain current can be obtained at lower gate bias. This is the reason why the OFETs here have an operating voltage of 3V. Similar low gate bias operation can be achieved with other dielectric/semiconductor combinations that give a high mobility. What is more interesting is that with a high mobility also higher currents can be induced, for example to drive an OLED. What is then important to know is how large the break down field (voltage over thickness) is, such that one can estimate what the maximal current is at which such an

OFET can operate. The break down field is not mentioned and also not compared to other dielectrics. This should be added.

Our reply: Thank you very much for your valuable suggestions. A sandwiched device structure of Au/insulator films/indium tin oxide (ITO)/PET was used to test the break-down field. With the same fabrication process but different annealing temperatures, the break-down field measurement of the polymers is shown in Figure 2b. PAA dielectric layer could only withstand less than 400 mV cm^{-1} electric field, while this copolymer film exhibited much higher break-down field (about 650 mV cm^{-1}), which is almost twice that of PAA and very close to fully cross-linked PI system (about 660 mV cm^{-1}). These results further confirmed that our copolymer design could indeed improve the insulating properties of the dielectric layer.

Our revision: In page 6, we added the revised contents. “Besides, PAA dielectric layer could only withstand less than 400 mV cm^{-1} electric field, while this copolymer film with same thickness exhibited much higher break-down field (about 650 mV cm^{-1}), which is almost twice that of PAA and very close to fully cross-linked PI system (about 660 mV cm^{-1}) (Fig. 2b)”

New figure:

Figure 2 | Current density, Break-down field, XRD, GIXRD and AFM characterizations. a) Current density of the dielectric layers at the bias voltage of 5V. Inset, an Au/Dielectrics (160 nm)/ITO/PET sandwiched device structure for test. (b) Current density as a function of electric field, (c) XRD patterns of pentacene films (50 nm) grown on copolymer and PI surfaces, (d) 2D GIXRD patterns of pentacene films (50 nm) on the surface of copolymer and PI. (e)- (j) AFM images of pentacene films grown on dielectric substrates with different thicknesses (4.5 nm, 15 nm and 50 nm): (e, g, i) copolymer; (f, h, j) PI.

Comment 3: Gate insulators often trap electrons due to for example OH groups, leading to p-type operation only. Other dielectrics also allow electron transport, leading to ambipolar transistors. Does the dielectric presented here also allow ambipolar operation?

Our reply: Thank you very much for your professional comments. We were also concerned about the issue that polar groups ($-\text{COOH}/-\text{CONH}$) might affect the electron transport before our experiments. However, during the measurement of the devices, we found that n-type semiconductor performed well on this copolymer dielectric (Supplementary Fig. 20). Further, based on your comments, we did more experiments to verify whether our copolymer allowed ambipolar operation. We are happy to see that the copolymer also allowed ambipolar operation and Supplementary Fig. 21 showed the typical transfer and output curves of the ambipolar transistors based on α , ω -Bis (biphenyl) terthiophene (BP3T). More related ambipolar research is in progress and we are looking forward to more potential applications of this copolymer.

Our revision: In page 11, we added the revised contents. “Moreover, this copolymer also allowed ambipolar operation and Supplementary Fig. 21 shows the typical transfer and output curve of this ambipolar transistor based on α , ω -Bis (biphenyl) terthiophene (BP3T).⁴⁶”

New figure:

Figure S21 | The performance of the BP3T based OTFTs. (a, c) Typical p-type transfer curve ($V_{DS}=-10V$) of the OTFT with BP3T as active layers. (b, d) Typical n-type transfer curve ($V_{DS}=10V$) of the OTFT with BP3T as active layers.

Thanks again for your encouraging comments and great contribution to our manuscript!

Sincerely yours,

Wenping Hu

Reviewer #1 (Remarks to the Author):

In the revised manuscript (NCOMMS-17-23269A), the authors performed additional experiments and analysis and provided more discussions on the comparison among OFET devices based on different dielectric materials.

My previous comments on the technical issues have been well addressed by the authors. The only issue left is the comparison between PI and PAA. I do not think the authors have provided evidences strong enough to convince me the superiority of PI in comparison to the more readily attainable PAA. Despite that the PI-based devices showed slightly better air stability (Figure S15a and S15b should have the same scale, i.e. both linear-scale or both log-scale), slightly reduced hysteresis and an enhanced breakdown voltage, the PI-based dielectric layer still shows much lower mobility than the PAA-based one.

I believe the current manuscript is notably improved over the previous one. The characterization/discussion are more complete and sound than the previous version.

Reviewer #2 (Remarks to the Author):

The authors have revised the manuscript based on the reviewers' comments. Several issues, however, still needed to be addressed as discussed below. The operating mechanism of the PAA/PI copolymer-based dielectric layer is still not clear, and this is the most critical issue of this manuscript. The paper thus is not ready for publication in Nature Communications at this stage.

Although the authors mentioned that the low-k dielectric materials can achieve low voltage operation by reducing the dielectric thickness. Compared to commonly used SiO₂, PAA/PI copolymer in this study has almost the same dielectric constant. Since SiO₂ cannot achieve an operation voltage lower than 3V with a dielectric thickness of 160 nm, the dielectric layer reported here may have a double-layer capacitor effect and that may explain why it could be operated at a low voltage. This needs to be carefully studied by testing the dielectric layer at low frequency (below 1 Hz). The authors only measured the capacitance as the frequency over 1000 Hz. If they do not have access to dielectric measurement at low frequency, they should prepare transistor devices with dielectric layer thicknesses. If the output current does not change with increasing dielectric layer thickness, it indicates that double-layer capacitor effect exist and the mobility reported will be over-estimated. In that case, the authors will need to characterize the dielectric layer capacitance at below 1Hz in order to accurately determine mobility.

For Referee 1

Dear Referee,

We greatly appreciated your encouraging and valuable suggestions for our manuscript (NCOMMS-17-23269A). According to your comments, we made corresponding revisions marked red in the manuscript.

Thanks again for your insightful comments!

Comment 1

In the revised manuscript (NCOMMS-17-23269A), the authors performed additional experiments and analysis and provided more discussions on the comparison among OFET devices based on different dielectric materials. My previous comments on the technical issues have been well addressed by the authors. The only issue left is the comparison between PI and PAA. I do not think the authors have provided evidences strong enough to convince me the superiority of PI in comparison to the more readily attainable PAA. Despite that the PI-based devices showed slightly better air stability (Figure S15a and S15b should have the same scale, i.e. both linear-scale or both log-scale), slightly reduced hysteresis and an enhanced breakdown voltage, the PI-based dielectric layer still shows much lower mobility than the PAA-based one. I believe the current manuscript is notably improved over the previous one. The characterization/discussion are more complete and sound than the previous version.

Our reply: First of all, thank you very much for your positive comments on the improvements of our manuscript!

In our previous work (Org. Electron. 2013, 14, 2528-2533), the biggest advantage of pure polyimide (PI) dielectrics is the excellent insulating properties due to high chain-packing density (100% imidization). However, the devices with pure PI dielectrics showed low mobility ($0.55 \text{ cm}^2 \text{ V}^{-1} \text{ s}^{-1}$). In addition, the high processing temperature ($300 \text{ }^\circ\text{C}$) of PI is not compatible with low-cost flexible substrates.

In comparison, although pure PAA dielectrics (J. Am. Chem. Soc. 2017, 139, 2734-2740) could enhance the mobility up to $30 \text{ cm}^2 \text{ V}^{-1} \text{ s}^{-1}$ because of the introduction of polar groups ($-\text{COOH}/-\text{CONH}$), the insulating properties of PAA dielectrics were apparently compromised. For example, the current density of PAA dielectric increased about two orders of magnitude (from $\sim 7 \times 10^{-10} \text{ A cm}^{-2}$ to $\sim 5 \times 10^{-7} \text{ A cm}^{-2}$, Figure 2a) compared with polyimide dielectric due to low chain packing density caused by lack of interaction between the phenyl rings and alicyclic rings. Besides, the main body of the PAA is acknowledged to be unstable and easy to degrade after long-time exposure to the air (J. Appl. Polym. Sci. 1985, 30, 2883), which results in the instability of the devices based on PAA dielectrics. As a result, 13% degradation of device performance (based on PAA dielectrics) was observed only after 60 days (Figure S16a).

The major novelty of the present work is to offer a simple and highly efficient strategy for the preparation of a new structure copolymer dielectric. This copolymer can well balance the insulating properties and charge transport for flexible organic thin-film transistors (OTFTs) applications. More interestingly, the two monomers produced with this method (see Scheme 1) play different roles: one could increase the chain-packing density guaranteeing sufficient insulating properties (e.g., the current density of the PAA dielectric is at least an order of magnitude higher than that of the copolymer, from $\sim 5 \times 10^{-7} \text{ A cm}^{-2}$ down to $\sim 1 \times 10^{-8} \text{ A cm}^{-2}$, Figure 2a) and the other monomer is responsible for surface polarity optimizing molecular packing of organic semiconductors. Benefitting from a good balance of the above two aspects, we see much higher performances (mobility up to $5.6 \text{ cm}^2 \text{ V}^{-1} \text{ s}^{-1}$, on/off current ratio of 1.4×10^6 , a threshold voltage of 0.42 V and a subthreshold swing of 220 mV/dec) of the OTFT devices with this copolymer dielectric than the previously reported results based on polyimide dielectrics

shown in Figure 5 and Table S2. From an application point of view, the mobility value and operation voltage are already good enough ($5.6 \text{ cm}^2 \text{ V}^{-1} \text{ s}^{-1}$ and 3 V operating voltage).

Compared with the devices using PAA dielectrics, copolymer-based devices exhibit comparable characteristics (such as operating voltage, on/off ratio, threshold voltage and subthreshold swing) and some better properties, including current density (Figure 2a) and hysteresis-effect (Figure S15). At the same time, PAA dielectric layers could only withstand electric fields of less than 400 mV cm^{-1} , while the copolymer film with same thickness exhibited much higher break-down field (about 650 mV cm^{-1}), which is almost twice that of PAA and very close to fully cross-linked PI system (about 660 mV cm^{-1}) (Figure 2b). More importantly, the stability was apparently enhanced by using this copolymer dielectrics: Only ~6% degradation of the device performance based on copolymer insulating layers was observed after 60 days compared to ~13% degradation of PAA-based devices). What's more, flexible organic field-effect transistors and circuits could also be successfully fabricated based on the copolymer insulators. Mobility is important for further applications (a decrease of the mobility of copolymer-based devices compared with that of PAA is ascribed to the different amount of polar groups, which was confirmed by the data shown in Figure S17), but the stability is even more vital for long-term development of organic electronics. In fact, the mobility of $5.6 \text{ cm}^2 \text{ V}^{-1} \text{ s}^{-1}$ in this study presents one of the best results of pentacene OTFTs to date and we believe it is very preferable for further practical applications in the future.

Figure S16 | The air-stability characterization of pentacene OTFTs based on copolymer and PAA dielectric. Mobility as a function of time (a) based on PAA (ref. 3), (b) based on copolymer.

Figure S17 | The characterization of PAA films under different annealing temperatures. (a) Attenuated total reflection (ATR) infrared spectroscopy of PAA with different imidization temperatures. (b) XRD patterns of pentacene films (50 nm)

grown on these three dielectrics. (c) Typical transfer curve of the OTFT with 50 nm pentacene and a channel dimension of $W = 240 \mu\text{m}$, $L = 30 \mu\text{m}$. (d) Mobility as a function of temperature.

Our revision: In page 10, we added the revised contents. “In addition, the devices showed outstanding operating stabilities in more than 4500 cycling tests of the transfer characteristics (**Supplementary Fig. 13**) and good environmental stability during shelf-life tests for more than 60 days (only 6% degradation of device performance was observed, **Supplementary Fig. 14**).”

New figure

Figure S16 | The air-stability characterization of pentacene OTFTs based on copolymer and PAA dielectric. Mobility as a function of time (a) based on PAA (ref. 3), (b) based on copolymer.

Thanks again for your great contribution to our manuscript!

Sincerely yours,

Wenping Hu

Referee 2

Dear Referee,

We greatly appreciate your insightful comments and suggestions for our manuscript (NCOMMS-17-23269A). In accordance with your suggestions, we have carried out more detailed investigations and obtained more positive supporting results, which help us further improve the quality of our manuscript. Our point-to-point replies and revisions (marked in red) are shown as follows. Thank you very much again for your insightful comments and great contribution to our manuscript!

Comment 1: The authors have revised the manuscript based on the reviewers' comments. Several issues, however, still needed to be addressed as discussed below. The operating mechanism of the PAA/PI copolymer-based dielectric layer is still not clear, and this is the most critical issue of this manuscript. The paper thus is not ready for publication in Nature Communications at this stage.

Our reply: Thank you for your valuable comments. To understand the interface electronic structures between copolymer and pentacene, we carried out in-situ thickness-dependent ultraviolet photoelectron spectroscopy (UPS) measurements. Figure 4a (new figure) shows the UPS spectra presenting the evolution of secondary electron cut-off (SECO) region and highest occupied molecular orbital (HOMO) region at the pentacene/copolymer interface. It is obvious that the vacuum level (VL) decreased gradually from 4.89 to 4.57 eV after in-situ incremental deposition of pentacene on the copolymer, which indicated the charge (electron) transfer from pentacene to copolymer upon contact. Meanwhile, the HOMO of pentacene shifts 0.18 eV towards the higher binding energy with its HOMO peak and leading edge from 0.67 and 0.24 eV to 0.85 and 0.42 eV below E_F . The derived schematic energy level diagram at pentacene/copolymer interface is depicted in Figure 4b, where the HOMO positions are directly derived from the UPS measurements, and the lowest unoccupied molecular orbital (LUMO) edges are estimated by adding the optical band gaps of 3.46 and 2.30 eV for copolymer and pentacene, respectively, to their corresponding HOMO energy level. It can be concluded that when pentacene and copolymer come into contact, electrons would move from pentacene to copolymer across the interface and holes would be created (left) in pentacene. Consequently, the accumulation of holes and electrons at

the interface leads to substantial band bending in both the pentacene and the copolymer layers. In a pentacene/copolymer OFET, the free holes in pentacene side can be easily driven along the interface by an electric field applied across the source and drain electrodes. Hence, the pentacene/copolymer interface is favorable for charge transport process.

Figure 4 | Interface characterization. (a) UPS spectra of incremental pentacene films on copolymer and (b) the derived energy-level diagram at the interface.

Also the role of polar groups ($-\text{COOH}/-\text{CONH}$) on the surface was further investigated. We decreased the annealing temperature of PAA films to 180 °C, 160 °C and 140 °C respectively for comparison. ATR infrared spectroscopy (Supplementary Fig. 17a) showed that with the decrease of the annealing temperature (< 200 °C), the imidization degree of the dielectric films dramatically reduced (46.44% for 180 °C, 41.43% for 160 °C and 25.41% for 140 °C), which meant that the density of polar groups ($-\text{COOH}/-\text{CONH}$) on the surface correspondingly increased. With the same fabrication process, pentacene films exhibited higher crystallinity on all the dielectric layers (Supplementary Fig. 17b). From typical transfer curves (Supplementary Fig. 17c), it can be observed that the existence of polar groups ($-\text{COOH}/-\text{CONH}$) on the surface enhanced the carrier transport and the mobility of the devices was inversely proportional to the imidization degree of the dielectric films (Supplementary Fig. 17d). Similar to PAA, the copolymer maintained surface polar groups ($-\text{COOH}/-\text{CONH}$) that could provide pronounced repulsive forces between the π -electron clouds of the pentacene backbone and the unshared

electron pairs of oxygen atoms in the COOH-functionalized dielectric, leading to more ordered packing and higher crystalline film with pentacene molecules standing on its surface. Therefore, the introduction of polar groups ($-\text{COOH}/-\text{CONH}$) plays a crucial role on the performance of this system.

The operating mechanism of the copolymer-based dielectric layer is thus summarized as follows:

(1) The accumulation of holes and electrons at the interface leads to substantial band bending in both the pentacene and the copolymer layers. In a pentacene/copolymer OFET, the free holes in pentacene side can be easily driven along the interface by an electric field applied across the source and drain electrodes. Hence, the pentacene/copolymer interface is favorable for charge transfer. (2) the copolymer maintained surface polar groups ($-\text{COOH}/-\text{CONH}$) that could provide pronounced repulsive forces between the π -electron clouds of pentacene backbone and the unshared electron pairs of oxygen atoms in the COOH-functionalized dielectric, leading to more ordered packing and higher crystalline film with pentacene molecules standing on its surface. The above-mentioned two aspects jointly guarantee the high performance of copolymer-based devices.

Figure S17 | The characterization of PAA films under different annealing temperatures. (a) Attenuated total reflection (ATR) infrared spectroscopy of PAA with different imidization temperature. (b) XRD patterns of pentacene films (50 nm) grown on these three dielectrics. (c) Typical transfer curves of the OTFTs with 50 nm pentacene and a channel dimension of $W = 240 \mu\text{m}$, $L = 30 \mu\text{m}$. (d) Mobility as a function of annealing temperature.

Our revision: In page 11, we added the revised contents. “To understand the interface electronic structures between copolymer and pentacene, we carried out in-situ thickness-dependent ultraviolet photoelectron spectroscopy (UPS) measurements. Figure 4a shows the UPS spectra presenting the evolution of secondary electron cut-off (SECO) region and highest occupied molecular orbital (HOMO) region at the pentacene/copolymer interface. It is obvious that the vacuum level (VL) decreased gradually from 4.89 to 4.57 eV after in-situ incremental deposition of pentacene on the copolymer, which indicated the charge (electron) transfer from pentacene to copolymer upon contact. Meanwhile, the HOMO of pentacene shifts 0.18 eV towards the higher binding energy with its HOMO peak and leading edge from 0.67 and 0.24 eV to 0.85 and 0.42 eV below E_F . The derived schematic energy level diagram at pentacene/copolymer interface is depicted in Figure 4b, where the HOMO positions are directly derived from the UPS measurements, and the lowest unoccupied molecular orbital (LUMO) edges are estimated by adding the optical band gaps of 3.46 and 2.30 eV for copolymer and pentacene, respectively, to their corresponding HOMO energy level. It can be summarized that when pentacene and copolymer come into contact, electrons would move from pentacene to copolymer across the interface and holes would be created (left) in pentacene. Consequently, the accumulation of holes and electrons at the interface leads to substantial band bending in both the pentacene and the copolymer layers. In a pentacene/copolymer OFET, the free holes in pentacene side can be easily driven along the interface by an electric field applied across the source and drain electrodes. Hence, the pentacene/copolymer interface is favorable for charge transport process.”

New figure

Figure 4 | Interface characterization. (a) UPS spectra of incremental pentacene films on copolymer and (b) the derived energy-level diagram at the interface.

Comment 2: Although the authors mentioned that the low- k dielectric materials can achieve low voltage operation by reducing the dielectric thickness. Compared to commonly used SiO_2 , PAA/PI copolymer in this study has almost the same dielectric constant. Since SiO_2 cannot achieve an operation voltage lower than 3V with a dielectric thickness of 160 nm, the dielectric layer reported here may have a double-layer capacitor effect and that may explain why it could be operated at a low voltage. This needs to be carefully studied by testing the dielectric layer at low frequency (below 1 Hz). The authors only measured the capacitance as the frequency over 1000 Hz. If they do not have access to dielectric measurement at low frequency, they should prepare transistor devices with dielectric layer thicknesses. If the output current does not change with increasing dielectric layer thickness, it indicates that double-layer capacitor effect exist and the mobility reported will be over-estimated. In that case, the authors will need to characterize the dielectric layer capacitance at below 1Hz in order to accurately determine mobility.

Our reply: Thank you very much for your suggestions! First of all, inorganic dielectrics and organic dielectrics cannot be directly compared, even with the similar dielectric constant. It has been well acknowledged that the defects on the top surface of SiO_2 are not well defined leading to interface trapping that may deteriorate the performance of the OTFTs (Chem. Mater. 2004, 16, 4543; Adv. Mater.

2005, 17, 1795; Adv. Mater. 2005, 17, 2411; Adv. Mater. 2008, 20, 2567). Therefore, to avoid predictable leakage current between the drain/source electrodes and the gate electrodes, the thickness of SiO₂ was normally controlled to be larger than 300 nm (Adv. Mater. 2012, 24, 4618; Adv. Mater. 2012, 24, 5735; J. Am. Chem. Soc. 2004, 126, 3378; J. Am. Chem. Soc. 2006, 128, 16002; J. Am. Chem. Soc. 2009, 131, 9396).

Furthermore, the interface trap density is an important factor. In our study, the subthreshold swing showed a low interface trap density of $3.7 \times 10^{11} \text{ cm}^{-2} \text{ eV}^{-1}$. For comparison, other three dielectric layers (160 nm PI, 50 nm SiO₂ and 300 nm SiO₂) were chosen for OTFT measurements to calculate their interface trap densities. 50 nm-thick pentacene was deposited on these insulators. From the curve of $\lg(-I_{\text{DS}})-V_{\text{GS}}$, the interface trap densities were $2.63 \times 10^{12} \text{ cm}^{-2} \text{ eV}^{-1}$ (PI), $1.65 \times 10^{13} \text{ cm}^{-2} \text{ eV}^{-1}$ (50 nm SiO₂) and $2.1 \times 10^{13} \text{ cm}^{-2} \text{ eV}^{-1}$ (300 nm SiO₂), respectively. As compared to the one-order and two-order of magnitude higher trap density in PI (160 nm) and SiO₂ (50 nm and 300 nm), respectively (Supplementary Fig. 9), the low interface trap density of copolymer indicated excellent dielectric-semiconductor interface quality and also demonstrated only a low gate voltage was required to attract holes to fill the charge trap states before accumulation occurring during operation of the OTFT (Adv. Mater. 2011, 23, 1630). In short, the low interface trap density and the highly ordered packing of pentacene molecules jointly contribute to the low operating voltage of the devices.

Figure S9| The measurements of the interface trap density. (a) 160 nm PI, (b) 50 nm SiO₂ and (c) 300 nm SiO₂.

Although there are two monomers in this copolymer backbone, there is only one single layer of copolymer dielectric. In order to further verify whether there exists double-layer capacitor effect in the

copolymer, we did the capacitance measurements both in the air and in the vacuum. Five different thicknesses (~ 250 nm, ~ 350 nm, ~ 650 nm, ~ 900 nm and ~ 1100 nm) of the copolymer films were tested both in the low-frequency and high-frequency region (from 20 Hz to 100 kHz, the instrument testing limit is 20 Hz). Supplementary Fig. 8 shows capacitance values as a function of the thickness of dielectrics. It was clear that there was little difference of capacitance in the air and in the vacuum, especially in the low-frequency region. Besides, we also fabricated transistor devices ($W/L=16$) with different dielectric layer thicknesses (~ 850 nm_(T4) > ~ 620 nm_(T3) > ~ 400 nm_(T2) > ~ 140 nm_(T1), Figure S23) and their transfer curves obviously changed as a function of thickness. All above experimental results exclude the possibility of double-layer capacitor effect in this system. What's more, all the mobility values were calculated at 20 Hz. In this way the mobility values we present in the paper are relatively conservative rather than overestimated.

Figure S8 | The frequency dependence of capacitance for copolymer dielectrics with different thickness under 200 °C annealing temperature. Negligible difference of capacitance was observed in the air and in the vacuum, especially in the low frequency region.

Figure S23 Typical transfer curves of the OTFTs with 50 nm pentacene deposited on copolymer dielectric layers with different thickness.

Our revision: In page 9 and page 10, we added the revised contents. “To more accurately calculate the dielectric constant, capacitance values of the copolymers with various thicknesses were tested both in the air and in the vacuum (from 20 Hz to 100 kHz) (Supplementary Fig. 8).”“ From Fig. 3c, the gate current was smaller than the drain current by more than five orders of magnitude, which further confirmed the high insulating property of this copolymer film. In addition, the subthreshold swing showed a low interface trap density⁴⁵ of $3.7 \times 10^{11} \text{ cm}^{-2} \text{ eV}^{-1}$. As compared to the one-order and two-order of magnitude higher trap density in PI (160 nm) and SiO₂ (50 nm and 300 nm), respectively (Supplementary Fig. 9), the low interface trap density of copolymer indicated excellent dielectric-semiconductor interface quality and also demonstrated only a low gate voltage was required to attract holes to fill the charge trap states before accumulation occurring during operation of the OTFT.⁴⁶ By using 160 nm-thick copolymer dielectric, the operating voltage was reduced to be as low as 3V, which was more than an order of magnitude smaller compared with previous reports, representing a big step forward towards practical application of polyimide-based OTFTs.”

New figures:

Figure S8 | The frequency dependence of capacitance for copolymer dielectrics with different thickness under 200 °C annealing temperature. Negligible difference of capacitance was observed in the air and in the vacuum, especially in the low frequency region.

Figure S9| The measurements of the interface trap density. (a) 160 nm PI, (b) 50 nm SiO₂ and (c) 300 nm SiO₂.

Thanks again for your great contribution to improve our manuscript!

Sincerely yours,

Wenping Hu

Reviewer #1 (Remarks to the Author):

The authors have revised the manuscript and provided the comparison with PAA dielectrics. In my opinion, the manuscript can be accepted now.

Reviewer #2 (Remarks to the Author):

The authors have addressed the comments from reviewer in the manuscript by adding deeper characterizations. In this revised version, the author introduced a paper from Dr. J.-C. Hwang (*Adv. Mater.* 23, 1630-1634 (2011)) to explain the role of those polar –COOH/–CONH groups in the dielectric layer. However, Hwang research group later also emphasized these kinds of dielectric materials with polar groups are easily affected by water (*Appl. Phys. Lett.* 103, 023303 (2013)). Even the author didn't observe the change of dielectric properties of 200 oC annealed film, it is still worth to analyzing the dielectric properties from those dielectric layers annealed at 140, 160, and 180 oC to clearly understand the dielectric property with polar groups. Also, from the new Figure S17, the transfer curve of 140 oC annealed device is not ideal (*Nat. Mater.* 17, 2-7 (2018)), and its mobility should be calculated by the capacitance of 140 oC annealed PAA/PI film. The authors need to provide all detail dielectric information. The capacitance measurement is suggested to be measured not only under vacuum and in air but also under different relative humidity (*Org. Electronics* 14, 1170-1176 (2013)) to confirm the stability of the PAA/PI layer. Thus, a major revision is recommended for this paper.

For Referee 2

Dear Referee,

We greatly appreciate your comments and suggestions for our manuscript (NCOMMS-17-23269B), according to which we have carried out more detailed measurements/characterizations (capacitance measurements in the air, in the vacuum and under different humidity on the dielectric layers annealed at different temperatures) to illustrate the robustness of our work. More positive supporting results were obtained, which help us further improve the quality of our manuscript. Our point-to-point replies and revisions (marked in red) are shown as follows. Thank you very much again for your great contribution to our manuscript!

Comment 1: The authors have addressed the comments from reviewer in the manuscript by adding deeper characterizations. In this revised version, the author introduced a paper from Dr. J.-C. Hwang (Adv. Mater. 23, 1630-1634 (2011)) to explain the role of those polar –COOH/–CONH groups in the dielectric layer. However, Hwang research group later also emphasized these kinds of dielectric materials with polar groups are easily affected by water (Appl. Phys. Lett. 103, 023303 (2013)). Even the author didn't observe the change of dielectric properties of 200 oC annealed film, it is still worth to analyzing the dielectric properties from those dielectric layers annealed at 140, 160, and 180 oC to clearly understand the dielectric property with polar groups. Also, from the new Figure S17, the transfer curve of 140 oC annealed device is not ideal (Nat. Mater. 17, 2-7 (2018)), and its mobility should be calculated by the capacitance of 140 oC annealed PAA/PI film. The authors need to provide all detail dielectric information. The capacitance measurement is suggested to be measured not only under vacuum and in air but also under different relative humidity (Org. Electronics 14, 1170-1176 (2013)) to confirm the stability of the PAA/PI layer. Thus, a major revision is recommended for this paper.

Our reply: Thank you for your valuable comments. The previous capacitance measurements of the copolymer films (annealing at 200 °C, five different thicknesses) both in the air and in the vacuum from 20 Hz to 100 kHz were carried out in Hong Kong in November 2017. At that time the relative humidity was about 80-90% (even in the lab). It was clear that there was little difference of capacitance in the air and in the vacuum (Figure S8), even in the low-frequency region, suggesting that the humidity had little effect on the copolymer film (annealing at 200 °C). This time we did more experiments to analyze the dielectric properties of the polymer layers annealed at different temperatures (140 °C, 160 °C and 180 °C) in Hong Kong under a relative humidity of ~60% (March). The new results (Figure S17, new figure) also indicated little difference of capacitance in the air and in the vacuum. In order to further verify the effect of humidity on dielectric properties, we also carried out measurements in Germany under a manual control of the relative humidity (40%, 60% and 80%). As shown in Figure S18 (new figure), the experimental results were consistent with that in Hong Kong. All the above measurements proved the copolymer dielectrics are fairly stable under different conditions and they are competent to fabricate robust electronics. Also, by following your suggestions, we recalculated the mobility values of the devices shown in Figure S19.

In addition, you mentioned a previous work (*Appl. Phys. Lett.* 103, 023303 (2013)) in which the device was easily affected by water and the performance apparently changed only after 20 min storage in air (the relative humidity was around 60%-70%). A key point is the dielectric material collagen hydrolysate (*Appl. Phys. Lett.* 103, 023303 (2013)) is water soluble. In comparison, our copolymer is not only water insoluble, but also insoluble in most of the commonly used organic solvents. What's more, the close interaction (confirmed by the UPS and XPS measurements, shown in Figure 4) between the copolymer and the pentacene could possibly further protect the interface from being influenced by the moisture or other impurities in air.

Figure S8 | The frequency dependence of capacitance for copolymer dielectrics with different thickness under 200 °C annealing temperature. Negligible difference of capacitance was observed in the air (relative humidity of 80%-90%) and in the vacuum, especially in the low frequency region.

Figure S17 | The frequency dependence of capacitance for copolymer dielectrics under different annealing temperatures. Negligible difference of capacitance was observed in the air (relative humidity of 60%) and in the vacuum.

Figure S18 | The capacitance measurement under a manual control of the relative humidity. Negligible difference of capacitance was observed under different relative humidity.

Figure S19 | The characterization of PAA films under different annealing temperatures. (a) Attenuated total reflection (ATR) infrared spectroscopy of PAA with different imidization temperature. (b) XRD patterns of pentacene films (50 nm) grown on these three dielectrics. (c) Typical transfer curve of the OTFT with 50 nm pentacene and a channel dimension of $W = 240 \mu\text{m}$, $L = 30 \mu\text{m}$. (d) Mobility as a function of temperature.

Figure 4 | Interface characterization. (a) UPS spectra of incremental pentacene films on copolymer and (b) the derived energy-level diagram at the interface.

Our revision: In page 9, we added relative revised contents. “To more accurately calculate the dielectric constant, capacitance values of the copolymers with various thicknesses were tested (from 20 Hz to 100 kHz) both in the air (under relative humidity around 80%-90%) and in the vacuum (Supplementary Fig. 8). It was clear that there was little difference of capacitance in the air and in the vacuum, even in the low frequency region, suggesting that the humidity had little effect on the copolymer film.”

In page 10, we added relative revised contents. “What’s more, the devices were stable in the atmosphere environment with little influence by the moisture or other impurities in air, as indicated by their almost the same performance in the air and in the vacuum (Supplementary Fig. 10). The robustness of the devices could be ascribed to the close interaction between the copolymer and pentacene keeping their interface from being affected by ambient conditions.³⁸”

In page 12-13, we added relative revised contents. “The capacitance of these copolymer layers were accordingly tested (from 20 Hz to 100 kHz) both in the air (under relative humidity of ~60%) and in the vacuum. Little difference of capacitance in the air and in the vacuum was observed (Supplementary Fig. 17 and Fig. 18), which demonstrated outstanding robustness of these copolymer layers under different

annealing treatments.” “From typical transfer curves (Supplementary Fig. 19c), it can be observed that the existence of polar groups ($-\text{COOH}/-\text{CONH}$) on the surface enhanced the carrier transport and the mobility of the devices ($C_{140^\circ\text{C}}$, 30 nF cm^{-2} ; $C_{160^\circ\text{C}}$, 24 nF cm^{-2} ; $C_{180^\circ\text{C}}$, 22 nF cm^{-2} from Supplementary Fig. 17) was inversely proportional to the imidization degree of the dielectric films (Supplementary Fig. 19d).”

New figure

Figure S17 | The frequency dependence of capacitance for copolymer dielectrics under different annealing temperatures. Negligible difference of capacitance was observed in the air (relative humidity of 60%) and in the vacuum.

Figure S18 | The capacitance measurement under a manual control of the relative humidity. Negligible difference of capacitance was observed under different relative humidity.

Thanks again for your great contribution to our manuscript!

Sincerely yours,

Wenping Hu

Reviewer #2 (Remarks to the Author):

The authors have been addressed the comments from the reviewer, and can be published in Nature Communication.